# Photochemically responsive polymer films enable tunable gliding flights

Jianfeng Yang [1], M. Ravi Shankar[2] & Hao Zeng [1]✉

Miniaturized passive fliers based on smart materials face challenges in precise control of shape-morphing for aerodynamics and contactless modulation of diverse gliding modes. Here, we present the optical control of gliding performances in azobenzene-crosslinked liquid crystal networks films through photochemical actuation, enabling reversible and bistable shape-morphing. First, an actuator film is integrated with additive constructs to form a rotating glider, inspired by the natural maple samara, surpassing natural counterparts in reversibly optical tuning of terminal velocity, rotational rate, and circling position. We demonstrate optical modulation dispersion of landing points for the photo-responsive microfliers indoors and outdoors. Secondly, we show the scalability of polymer film geometry for miniature gliders with similar light tunability. Thirdly, we extend the material platform to other three gliding modes: Javan cucumber seed-like glider, parachute and artificial dandelion seed. The findings pave the way for distributed microflier with contactless flight dynamics control.

Stimuli-deformable materials, as a class of innovative substances, possess the unique ability to undergo programmable shape changes in response to specific external stimuli, such as heat, humidity, light, etc.[1–4]. In the realm of soft robotics, these materials hold great potential. Unlike traditional rigid robots, a single soft material actuator can morph and alter its shape through contracting, bending, or twisting modes[5–7], allowing for multiple degrees of freedom in robotic operation and remote task execution[8–10]. In this context, the advancement of miniature robots, capable of versatile walking on diverse surfaces and environments[11,12], swimming within narrow passages like blood vessels[13], or executing jumps[14], has sparked a wave of enthusiasm and pursuit for the next big challenge in the scope of micro-robotics research, that is, the flying robot.

To achieve a hovering flying robot (active flight mode) with dimensions in the centimeter scale and weight of hundreds of milligrams, it is essential for the actuating materials to possess high power density and high bandwidth[15]. However, this necessitates the utilization of piezoelectric and dielectric materials that rely on electrical energy transfer, requiring either cables or onboard batteries[16,17]. For stimuli (e.g., light) field-deformed materials, despite their potential advantages in miniaturization and wireless control[18], there are certain limitations that render them unsuitable for hovering robots. One such limitation is their inability to provide the required power output for sustained hovering without compromising other essential functionalities[19]. Additionally, the soft materials may exhibit instability in their response during flight, which could impact the lift generation[20]. Thus, researchers have begun exploring alternative flight options that offer favorability in energy consumption.

Dispersal, as a strategy of continuous natural selection, is a crucial process that promotes the distribution and colonization of plant species[21]. In the natural kingdom, many seeds have evolved to utilize wind-assisted passive flight mechanisms effectively, allowing them to disperse to different locations. These mechanisms can be roughly divided into three categories[22]: (1) Seeds with a parachute-like structure, such as dandelions, have a filament structure that creates air resistance; (2) Seeds that can glide in the air, such as Javan cucumber that have wing-like structures, and (3) maple samara and similar species can autorotate in the air, generating lift through a helicopter-like mechanism. Extensive aerodynamic studies of these mechanisms have been conducted, drawing inspiration from natural examples[23], and pioneering studies about artificial dispersers that work as passive fliers[24–26]. Today, artificial dandelions based on smart materials not

[1]Light Robots, Faculty of Engineering and Natural Sciences, Tampere University, P.O. Box 541 Tampere, Finland. [2]Department of Industrial Engineering, Swanson School of Engineering, University of Pittsburgh, Pittsburgh, PA, USA. ✉e-mail: hao.zeng@tuni.fi

only exhibit similar properties of separated vortex ring generation and dispersing stably in the air like natural ones but are also capable of light-controllable take-off and landing in response to external stimuli[27]. However, the fragility of the pappus of dandelions limits their load-carrying capacity[27,28], posing challenges for certain propagations that require transporting heavier payloads or operating in harsh environments. For a quantitative comparison of load-carrying between helicopter-like and parachute-like fliers, see Supplementary Fig. 1.

The gliding mechanism, with robust structure and higher loading capacity, has sparked the development of bionic rotary passive fliers across different length scales[29,30], and the creation of light-vapor-powered active robots capable of rotary flying movements while airborne[31]. Besides the aforementioned advantages, current glider-inspired research has also revealed its limitation in the inability to modify aerial gliding performance[32]. A recent development has successfully addressed several important issues in gliding robots[33], i.e., fast actuation, self-powering, and control of the geometry mid-flight to modulate the aerodynamics while accessing functionalities using onboard sensors and wireless communication. Such an approach is based on a complex integration of multiple electronic elements within a miniaturized origami parachute. Alternately, optical shape morphing polymers set up an intriguing opportunity for controlling responsiveness from stand-off distances, contactlessly. Thus, setting up the question in the context of microfliers: Can light be a direct way to reconfigure the wing geometry and provide dynamic control of the

gliding performance? Can a single piece of polymer be used to modulate responsiveness, instead of an integrated electromechanical system built on multiple materials? Furthermore, can this shape-morphing allow for facile tuning of the gliding to elicit a range of modalities and abilities to steer the trajectory?

Here, we attempt to explore the aforementioned scientific questions by reporting light-deformable polymers that exhibit optically tuning in gliding performances in the air. It has reversible variations in altitude and rotational speed by altering the illumination between ultraviolet (UV) and visible light (460 nm). Additionally, we also demonstrate the opportunity for miniaturization, as well as the generalization of the concept to other gliding modes.

## Results

### An artificial rotary gliding seed

Maple samara, like other winged seeds (e.g., those of ash and tulip trees), utilize natural winds, i.e., updrafts or crosswinds for dispersal, enabling them to travel distances of several hundred meters or more[34]. The typical structure of a maple samara is shown in Fig. 1a, where vascular bundles are concentrated near the leading edge of the maple samara, forming a thick edge and a thin wing plane. A heavy seed root grows at the base of the maple samara, shifting the gravitational center position near the root. This naturally optimized chordwise mass distribution causes the maple samara to autorotate in the air. During the steady descent in the still air, the gravitational force is balanced by the

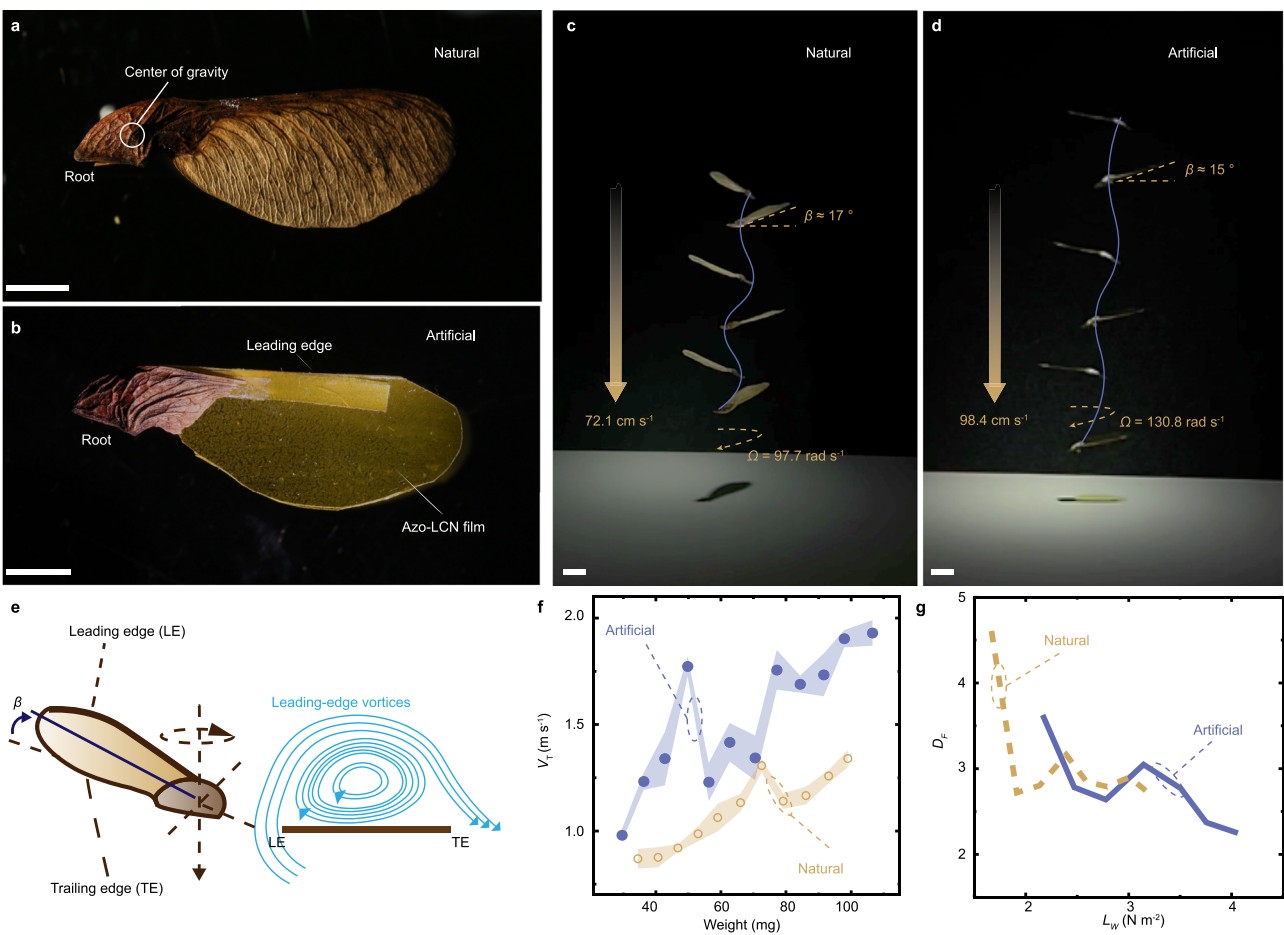

**Fig. 1 | An artificial rotary gliding seed.** Photographs of natural maple samara (**a**) and artificial seed (**b**). Scale bars are 5 mm. Superimposed images of natural maple samara (**c**) and artificial seed(**d**) during the descent in the steady air. Scale bars are 1 cm. **e** Schematic drawing of the airflow around a cross-section of the maple samara, and the formation of a leading-edge vortex. Reproduced with permission[22].

 **f** The measured terminal velocity ($V_T$) upon different loading conditions in both natural maple samara and artificial seeds. **g** Descent factor ($D_F$) at different wing loads ($L_w$). The error bars are displayed as mean values +/- standard deviation (n = 3). The same sample was measured repeatedly.

lift generated from the rotating wing. When air flows from the high-pressure region beneath the wing to the low-pressure region on top, it wraps around the leading edge, creating a swirling motion of air and forming a leading-edge vortex (LEV)[35], which will delay the separation of the airflow from the upper surface of the wing and contributes to lifting generation[36] (Fig. 1e). These features allow maple samara to possess merits over other natural seeds, including (i) a stable auto-rotary (helicopter-like) flight pattern and (ii) a high load capacity, which is 4 times higher than that of other gliding seeds (i.e., Alsomitra macrocarpa)[35].

Taking inspiration from maple samara, we fabricated a photo-deformable maple samara by assembling a soft actuator film, a natural seed root and an additive mass chord, as shown in the photograph in Fig. 1b. The soft actuator film is of a liquid crystal network (LCN)[37–39] crosslinked by azobenzene photoswitches[40–43]. This azo-LCN film provides reversible, bistable photochemical deformation and fine-tuning of aerodynamic properties, as elaborated in the later sections. The actuator film is trimmed following the contour of a maple samara. A natural seed root and an LCN strip are attached to the film end position and leading edge, respectively, to optimize the mass distribution on the wing plane following the natural design strategy. For the details of fabrication, see Supplementary Fig. 2 and Methods. Eventually, the components are assembled into an artificial seed that has a similar mass (33.1 mg, compared to a dry maple samara: 34 mg, in this study) and gliding properties as the natural counterpart. As depicted in Fig. 1c, d, during the free-fall, both artificial seeds and natural samaras exhibit autorotational behavior in the air (Supplementary Movie 1). The natural maple samara descends at a speed of 72.1 cm s$^{-1}$ with a cone angle $\beta$ about 17°, while the artificial seed shows a speed of 98.4 cm s$^{-1}$ and a $\beta$ about 15°. The similarity in cone angle $\beta$ indicates a comparable equilibrium between the centrifugal force acting on the distributed mass of the seed, the gravity of the seed, and the aerodynamic forces in both natural and artificial cases during their rotary gliding flight. For the force analysis, see the schematical drawing in Supplementary Fig. 3.

By adding weight to the roots of the autorotary seeds enables us to test their load-bearing capacities. As the weight increases, the terminal descent velocity $V_T$ and spinning rate $\Omega$ both increase (Fig. 1f). The descent factor $D_F = W/(\rho S V_T^2)$ is used as a measure of aerodynamic efficiency, where $\rho$ is the density of air, $W$ is the mass of seed, $S$ is the projected area, and the wing load $L_w = W/S$. By adding weight to the structure, we can plot the $V_T$ − weight data for both natural and artificial seeds (Supplementary Fig. 4). As shown in Fig. 1g, artificial seeds exhibit comparable aerodynamic efficiency to that of natural maple samaras upon different loading conditions, indicating that such a facilely assembled structure is able to qualitatively imitate the aerodynamics of the natural autorotary seeds.

## Photochemical and light tuning in artificial rotary seeds
What makes the artificial seed surpass the natural one is the photo-responsive wing capable of light-tuned mechanical deformation. To induce deformation, an azo-LCN film actuator is utilized for the wing construction. Prepared via photo-polymerization, this azo-LCN is crosslinked with azobenzene switches, which undergo trans-cis isomerization under UV irradiation and return to trans state upon exposure to visible light (vis). For details of film preparation, see Fig. 2a, Supplementary Fig. 5 and Methods. The isomerization takes place in the solid form, as evident in the UV-Vis spectra measured from a polymer film shown in Fig. 2b, where the maximum absorption peak shifts between 360 nm (corresponding to trans-azo absorption) to 460 nm (cis) by altering illuminating light sources between UV and visible light. The light causes a kinetic change in the population of trans and cis isomers inside the material that highly relies on the irradiating intensity. As shown in Fig. 2c, the trans isomer decreases from nearly 100% to <50% after 120 s upon dim light (360 nm, 4 mW cm$^{-2}$) and drops quickly upon strong irradiation, to 50% within 10 s upon

155 mW cm$^{-2}$ illumination. The change in configuration at the molecular level is transformed into macroscopic scale through azo-LCN with preserved splayed alignment, which yielding uni-directional bending independent of incident direction. Correlated with the change of absorbance spectra, such photochemical deformation exhibits similar kinetics that highly relies on light intensity (Fig. 2i). It is worth noting that the photo-induced deformation is yielded by pho-tochemical effect, under nearly isothermal conditions during the sample deformation (Supplementary Fig. 6). This provides merits in term of energy consumption and advantages for long period opera-tion. Compared to the light-thermal actuation that cannot efficiently alter the structural morphing due to the heat dissipation inside the wind and causes deformation reversal immediately after ceasing the light, photochemical LCN is capable of bistable shape-morphing. Supplementary Fig. 7 and Supplementary Table 1 shows the cis-lifetime about 300 minutes at room temperature, guaranteeing bistable shape-morphing (Supplementary Movie 2). The bistable actuation brings one benefit for the airborne robots: the deformed structure requires only one-shot UV light illumination for shape-morphing, after which it can maintain the fixed geometry during the entire flight until encountering another manipulated light field (visible light) for shape change.

As shown in Fig. 2d−f and Supplementary Movie 3, the photo-chemical actuation influences the autorotation and descent of the artificial seed in a reversible manner. After exposure to UV light, the deflection angle $\alpha$ of the wing decreases, as depicted by the side view photographs of artificial seed in Fig. 2g, h, and the tip displacement ($d$) data in Fig. 2i. Top view photographs of artificial seed before and after photo-actuation, see in Supplementary Fig. 8. The deflection results in a flatten wing plate and a reduced terminal descent velo-city $V_T$ from 98.4 cm s$^{-1}$ to 88.2 cm s$^{-1}$. The light's influence on the $V_T$ and $\Omega$ (spinning rate) is plotted in Fig. 2j as a function of the UV dose (intensity × exposure time). With an increase of UV dose, both the $V_T$ and $\Omega$ drop, indicating a prolonged airborne period (higher aerodynamic efficiency). Upon further increase in UV irradiation above $30 \times 10^3$ J m$^{-2}$, the film bends to the opposite side with a negative $\alpha$, and meanwhile, both $V_T$ and $\Omega$ start increasing. The reason behind this is an increase of $D_F$ (descent factor), which is associated with the decrease in deflection angle that can be tuned by the light dose (Fig. 2k). A change in the wing deflection angle (about 20°, Fig. 2j) can cause a significant alteration in gliding performance, implying that photochemical actuation could provide control of gliding performance in rotary dispersers (Fig. 3).

## Light-tuned rotary motion in the wind tunnel
To study the impact of light on the rotary behavior in air, an artificial seed is positioned above a vertical wind tunnel with electronic control in wind velocity. For the construction of wind tunnel, see "Methods" and the drawing in Supplementary Fig. 9. To center the glider inside the wind flow, a long fabric fiber (<20 μm in diameter) is threaded through a hole near the seed's mass center (Fig. 3a). The thread is fixed vertically, connecting the central spot of the tunnel output to a sup-portive cantilever above, confining the rotary motion on the horizontal plane and restricting system to nearly one degree of freedom, this means the rotary glider can either move up or down with minimal vibration in the lateral directions. A wind velocity gradient exists at the tunnel output, as the upward wind speed gradually decreases from the honeycomb plane to upper aerospace (see Supplementary Fig. 10). The seed autorotates within the wind tunnel and self-stabilizes at a certain height where the strength of wind flow allows generation of airlift that exactly balances the glider's gravitational force. Thus, by changing the flow velocity, the glider exhibits different stabilized heights and spinning rates (Fig. 3b, c).

Figure 3d shows the photographs of two artificial seeds inside a steady tunnel (output speed: 1.1 m s$^{-1}$). The seed on the right is capable of light tuning, while the left one is integrated with an undeformable

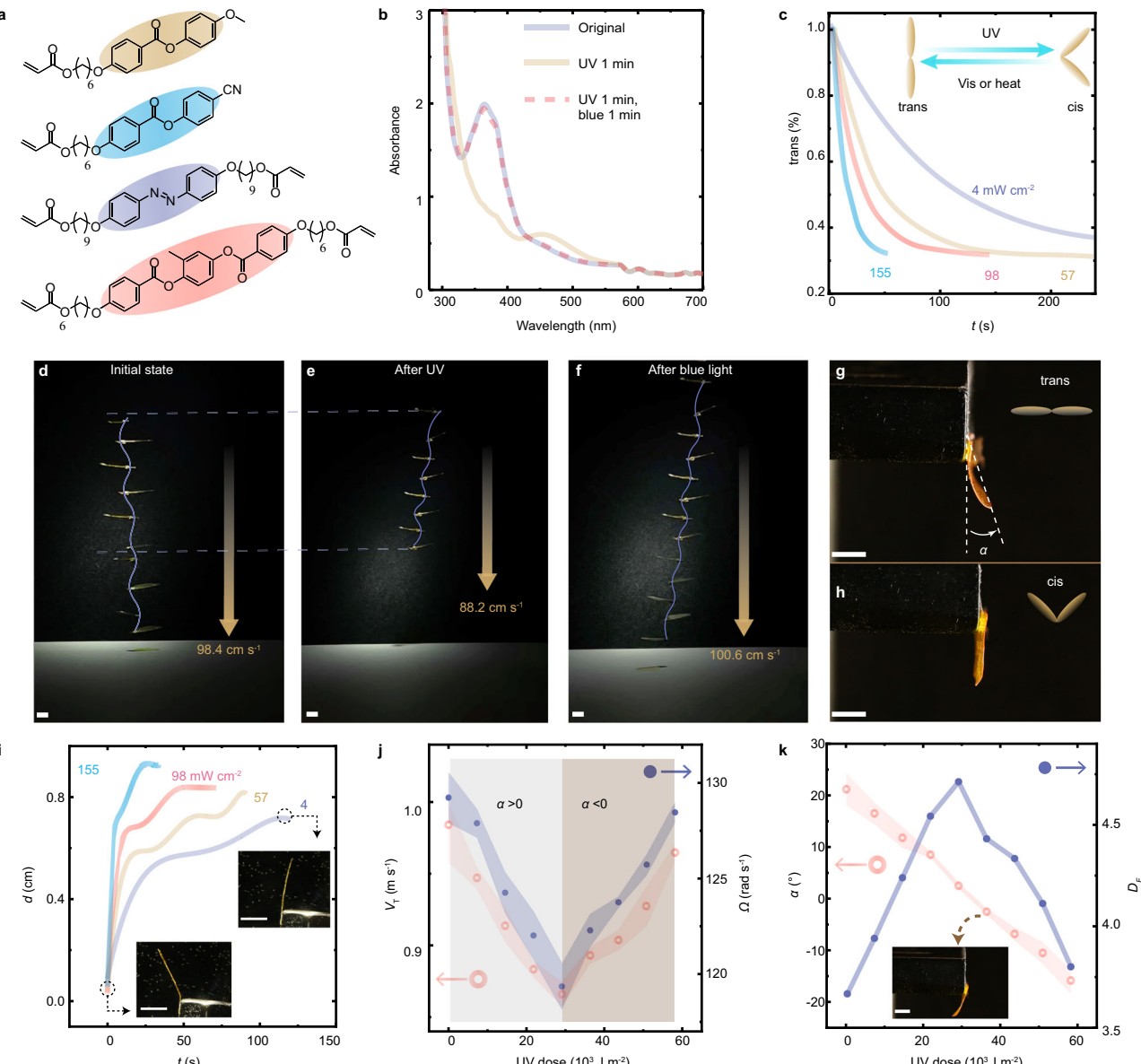

**Fig. 2 | Photochemical actuation and light tuning in artificial seeds. a** Chemical composition of the photomechanical actuator. **b** UV–Vis spectra of a solid actuator film. The thickness of the film: 5 μm. Alignment: splayed. **c** The change of trans-isomer content in the azo-LCN film upon irradiation of different UV intensities. Superimposed images of a free-falling artificial seed as an initial state (**d**), after UV (240 mW cm⁻², 12 s) illumination (**e**) and then after blue light (300 mW cm⁻², 20 s) irradiation (**f**). The shape of the artificial seed wing before (**g**) and after (**h**) UV irradiation. UV: 240 mW cm⁻², 12 s. **i** Deformation kinetics of the azo-LCN film under different UV intensities, insets: photographs of azo-LCN strip bending after UV illumination. **j** Change of terminal velocity ($V_T$, red ring) and spinning rate ($\Omega$, blue dot) of the artificial seed after receiving different UV doses. **k** Change of the deflection angle $\alpha$ (red ring) of the wing and corresponding descent factor ($D_F$, blue dot) upon different doses of UV irradiation. The error bars are displayed as mean values +/- standard deviation (n = 3). The same sample was measured repeatedly. UV wavelength: 360 nm. After each experiment, the film is illuminated by blue light (460 nm, 300 mW cm⁻², 30 s) for complete shape recovery. All scale bars are 5 mm.

LCN wing (polydomain sample) as control. Upon continuous UV irradiation, the height of the control sample remains unchanged, while the photo-responsive one raises its height. The kinetics of object height ($h$) and $\Omega$ changes are shown in Fig. 3e. UV illumination produces an increase of $D_F$ (as measured in Fig. 2k), resulting in higher aerodynamic efficiency for the rotary gliding. This yields a fact that the same equilibrium condition (gravitational force balanced by the airlift) necessitates a lower rotary rate and wind speed (at larger $h$). This is qualitatively explained by data shown in Fig. 3f, where the $h$ and $\Omega$ variations exhibit distinct trends upon irradiation. The maximum change of height $\Delta h_m$ increases along light intensity, and the same trend is observed in flight upon wind flow with different velocities (Supplementary Fig. 11).

The reversible trans-cis-trans isomerization of azobenzene molecules enables the reversed tuning of gliding behavior. Figure 3g shows the snapshots of the artificial seed moving upward and downward by altering the environmental illumination between UV and visible light sources (also see Supplementary Movie 4). Upon visible light illumination, the wing deflects backward and increases $\alpha$, yielding a decrease in $D_F$ and a reduction of $h$. The light tuning in the rotary gliding can be repeated for cycles, as shown by the $h$ and $\Omega$ variation upon five cyclic UV-Visible irradiations (Fig. 3h). Supplementary Fig. 12 shows the data of 25 cyclic light tunings of gliding performance inside the wind tunnel, and 100 cycles of photomechanical deformations in an actuator strip. Within the experimental time span, no decay in material capability is observed.

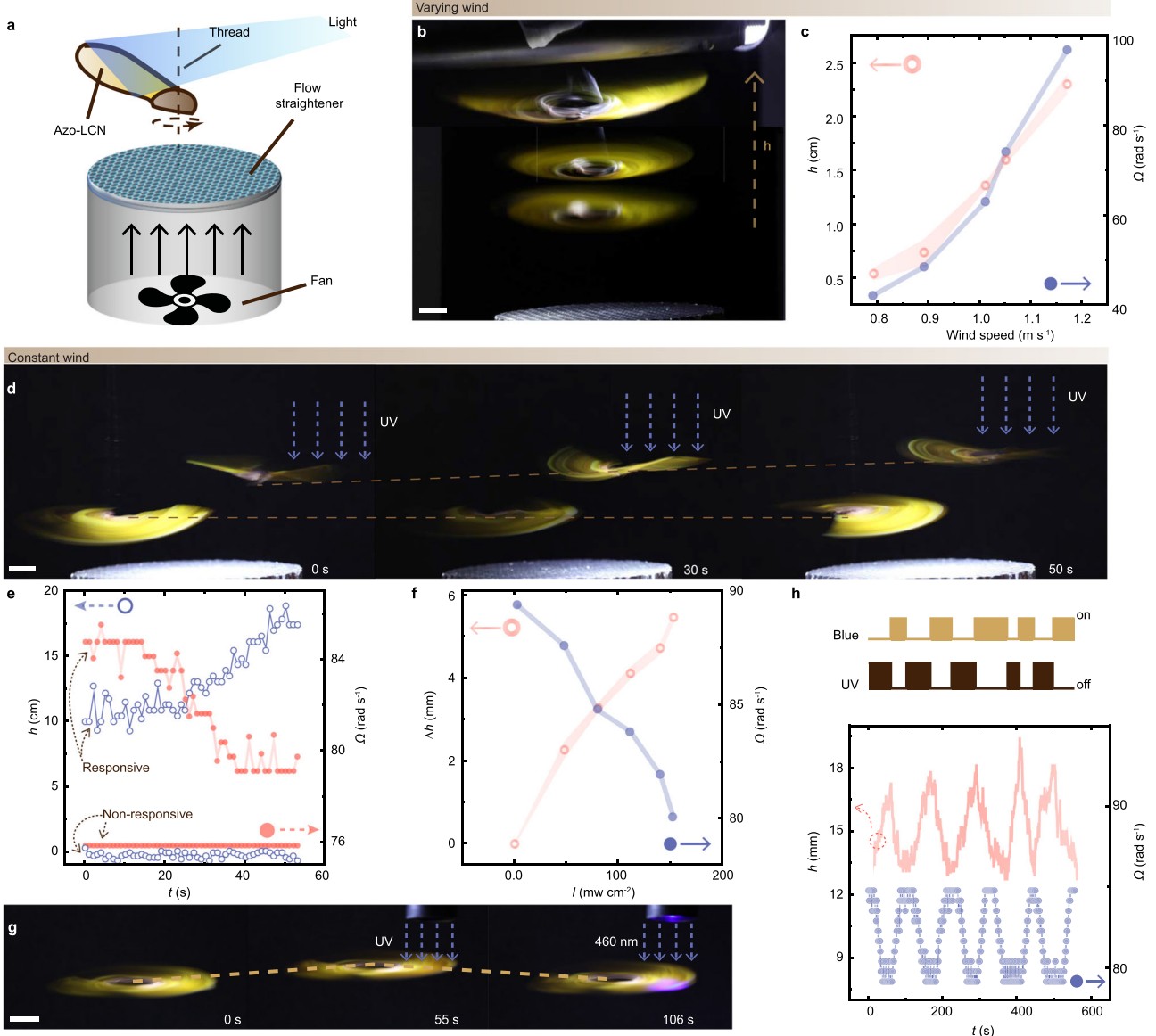

**Fig. 3 | Photo-tuning in autorotation behavior inside the wind tunnel.**
**a** Schematic drawings of the rotary seed inside the vertical wind tunnel.
**b** Superimposed picture of the height changes of the artificial seed by increasing upward wind flow from 0.8 m s⁻¹ to 1.2 m s⁻¹. **c** The changes of the height ($h$, red ring) and spinning rate ($\Omega$, blue dot) of the artificial seed by increasing the wind speed. The error bars are displayed as mean values +/- standard deviation ($n = 3$). The same sample was measured repeatedly. **d** Snapshots of height changes of a non-responsive (left) seed and a responsive (right) one upon UV exposure (150 mW cm⁻²). **e** The changes of $h$ (blue ring) and $\Omega$ (red dot) of non-responsive

and responsive seeds upon 150 mW cm⁻² UV excitation. **f** The maximal recorded changes of the height $\Delta h_m$ (red ring) and the spinning rate $\Omega$ (blue dot) of the artificial seed at different UV irradiation intensities. The error bars are displayed as mean values +/- standard deviation ($n = 3$). The same sample was measured repeatedly. The wind speed is 1.1 m s⁻¹. **g** The snapshot images of an artificial seed with reversible height change by altering the illumination between UV and visible light sources. UV: 360 nm, 150 mW cm⁻²; visible light: 460 nm, 400 mW cm⁻². **h** The changes of $h$ (red line) and $\Omega$ (blue dot) of the artificial seed upon changing the excitation wavelength. All scale bars are 1 cm.

## Light-tuned wind-dispersal

The following study demonstrates the control of airborne gliding behavior over a longer distance and the possibility of utilization in an outdoor environment. We first conduct free-fall experiments with artificial seeds irradiated by UV. The UV is illuminated with 20 s to ensure maximal trans-to-cis conversion and saturation of deformation (Fig. 2c). The seeds are placed at an identical release point of 2 m above the ground inside static air. About 1 m below the releasing point, there is a visible light zone (460 nm, 760 mW cm⁻²), as schematically shown in Fig. 4a. The visible light field relaxes the wing to its original state (cis-to-trans in azo-LCN) and changes descent velocity in the mid-air. Figure 4b shows vertical trajectories of four free-falling samples. The seeds are first accelerated under

gravitational force and reach aerodynamic equilibrium through spinning after half a meter of falling. After entering the light zone, all samples show an enhancement of descent speed. The $\Delta V_T$ is about 17% of the $V_T$ of non-UV exposure, which is consistent with previous examination data shown in Fig. 2d-f, j.

Secondly, we conduct multiple free-fall experiments involving both natural and light-irradiated artificial seed structures within a crosswind flow. The seeds are placed at an identical release point of 2 m above the ground near a horizontal wind fan (1 m height, flow velocity: $0.93 \pm 0.07$ m s⁻¹), as schematically shown in Fig. 4c. Details of setup, see "Methods" and Supplementary Fig. 13. All the seeds reached stabilized autorotation with steady descent velocity before entering the wind zone, which induced dispersal in the lateral directions.

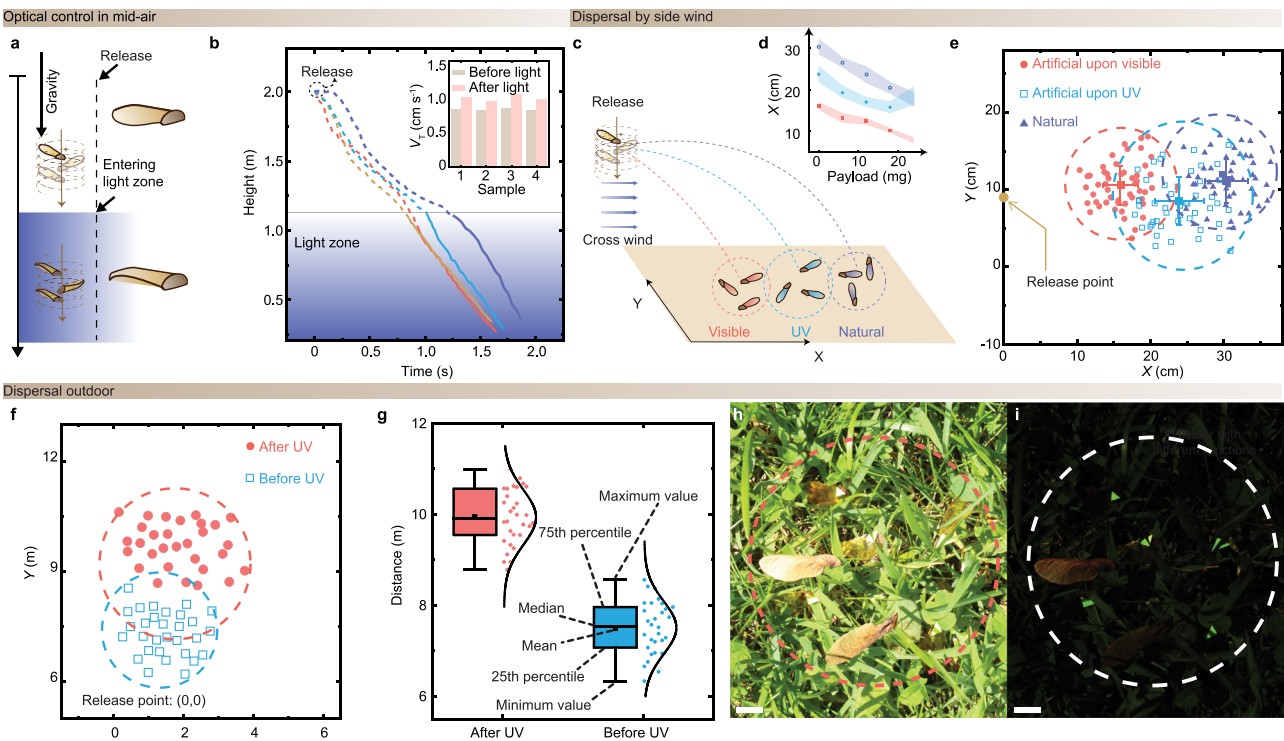

**Fig. 4 | Light-tuned wind-dispersal. a** Schematic drawings of light induced shape-morphing of artificial seeds in mid-air. **b** Free falling trajectories of artificial seeds across the visible light zone. Light: 460 nm, 760 mW cm⁻². Inset: The measured terminal velocities ($V_T$) of artificial seeds before (brown bars) and after entering (red bars) the visible light zone. Measurements were taken from distinct samples. **c** Schematic drawings of the free-fall experiments of gliding flights of artificial seeds with the help of a crosswind indoors. **d** The dispersal distance of natural maple samaras (purple), artificial seeds after the UV irradiation (blue, 150 mW cm⁻², 20 s) and artificial seeds after blue light illumination (red, 300 mW cm⁻², 30 s) upon different loadings. The error bars are displayed as mean values +/- standard deviation ($n = 3$). The same sample was measured repeatedly. Wind speed:

$0.93 \pm 0.07$ m s⁻¹. **e** The landing spot distribution of natural maple samaras (purple triangle), artificial seeds after the UV (blue square, 150 mW cm⁻², 20 s) and artificial seeds with the blue light irradiation (red dot, 300 mW cm⁻², 30 s) on the X-Y plane indoors. Wind speed: $0.93 \pm 0.07$ m s⁻¹. The same sample was measured repeatedly. **f** The landing spot distribution of artificial seeds before (red dot) and after (blue square) UV illumination in outdoor conditions. Height of the building: 14 m, wind speed at the releasing point: $4 \pm 0.5$ m s⁻¹. UV: 150 mW cm⁻², 20 s. **g** Statistics of landing point distance for artificial seeds after UV irradiation (red) and the same seeds before the UV irradiation (blue). UV: 150 mW cm⁻², 20 s. Measurements were taken from distinct samples. Day (**h**) and night (**i**) photos of artificial seeds equipped with light reflectors. All scale bars are 1 cm.

Figure 4e shows that the unilluminated artificial seeds travel at the shortest (15 cm, mean) due to their lowest $D_F$. After subjecting to UV (150 mW cm⁻², 20 s), the artificial seed exhibits an increased traveling distance (24 cm, mean). In comparison, natural maple samaras travel at the longest distance (30 cm, mean) under the conditions of study. This is due to the fact that natural maple samaras possess the lowest descent speed and, thus, the longest airborne duration. The statical data of the scattered land points (Supplementary Fig. 14) indicates the optical impact on the dispersal distance over space.

Thirdly, we conduct multiple free-fall experiments of artificial seeds in outdoor conditions. The seeds are released from a ladder outside of a building, which is 14 m above the ground. The instantaneous wind speed is measured to be about 4 m s⁻¹ at the releasing point. A photograph of the experimental location is shown in Supplementary Fig. 15. Figure 4f shows that the unilluminated artificial seeds land at a horizontal distance of about 7 m away from the releasing spot. The UV-illuminated samples can travel at an increased distance. Thus, a 3 m mean difference is observed between two sets of samples based on statistics from 30 experiments (Fig. 4g, Supplementary Fig. 16). As shown in Fig. 4d, as the loading weight increases, the dispersal distance drops, indicating the ability to modulate the aerodynamics, while sustaining a payload up to tens of mg. To enable their tracking, we incorporated the artificial seed with an optical indicator. The indicator film can reflect light directionally with large portion of back reflection along the reversed direction of incident

light. This allowed their utilization in non-laboratory conditions over extended flight paths (Fig. 4h, i). Other stimuli-responsive polymer films, i.e., pH indicator and humidity-responsive polymer, can also be incorporated into the seed body. This assembly allows for optical monitoring of the environmental conditions by tracking the colors of the responsive elements, which constitute the payload of the microfliers. Details can be found in Supplementary Fig. 17, while the sample preparation of humidity-responsive polymer is outlined in Supplementary Fig. 18 and the Supplementary Methods.

## Scaling down and conceptual generalization

The photochemical actuation in azo-LCN offers an opportunity to miniaturize the flier. As shown in Fig. 5a, we fabricated different sizes (3, 1.5, 0.75 and 0.3 cm) using a laser cutter. All sizes of artificial seeds achieve mid-air autorotation, with their descent speeds averaging around 100 cm s⁻¹. As depicted in Fig. 5b, after exposure to UV light, the terminal velocity of all sizes of artificial seeds decreased due to a reduction in the deflection angle $\alpha$ of the wing. Superimposed images of the light tunable seeds during the free-falling are shown in Fig. 5c, d, and Supplementary Fig. 19. The results are consistent with that observed in Fig. 2.

The shape-morphing azo-LCN also provides the opportunity to extend the concept to various passive flight modes. In nature, alongside those capable of autorotation like the maple samara, there exist seeds with sail-like wings for gliding (e.g., Javan cucumber) and seeds

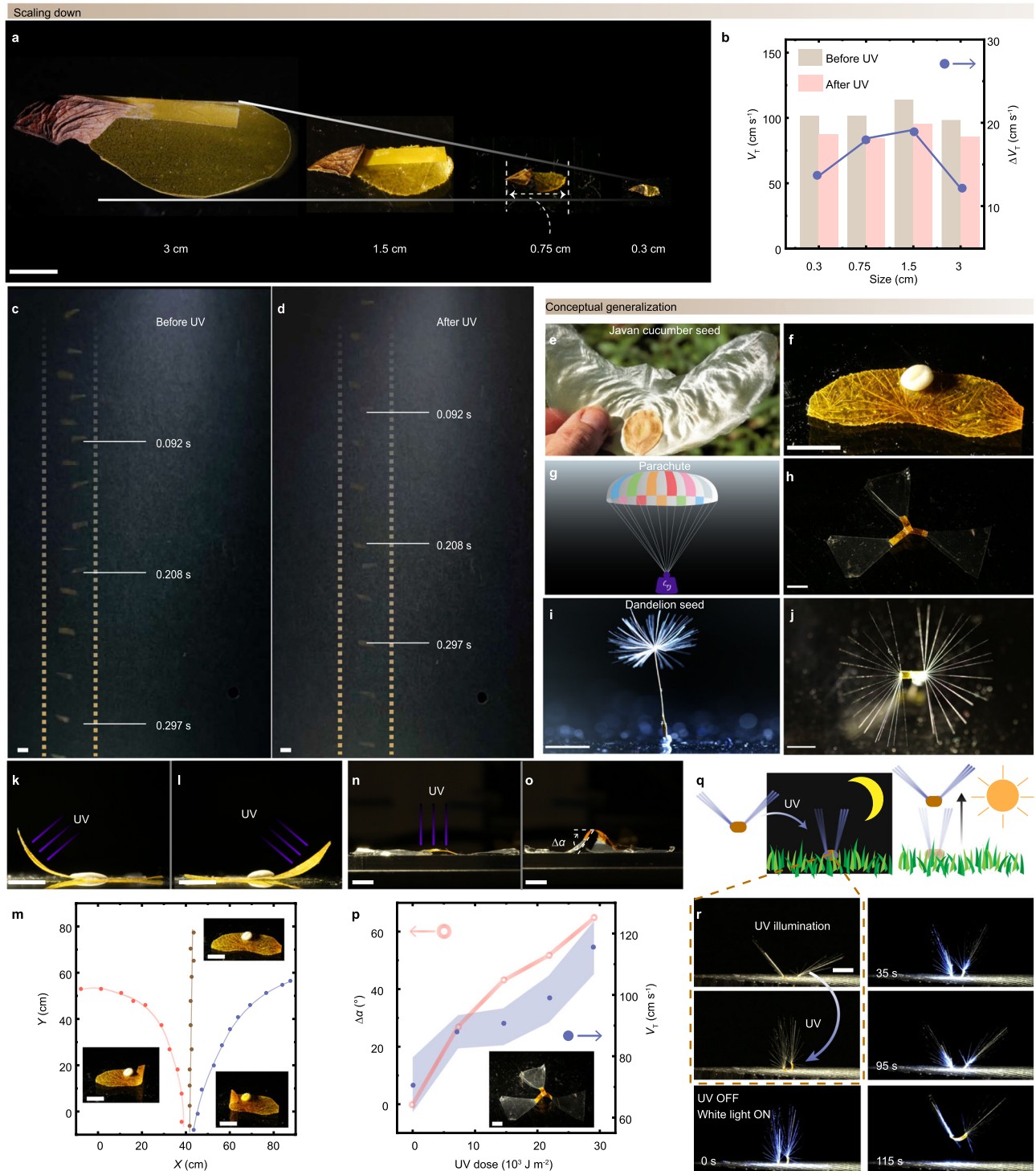

**Fig. 5 | Scaling down and conceptual generalization. a** Photographs of artificial seeds with sizes of 3, 1.5, 0.75, and 0.3 cm. **b** The measured terminal velocity ($V_T$) and the change of terminal velocity ($\Delta V_T$) before and after UV illumination in the seeds with different sizes. UV illumination: 150 mW cm$^{-2}$, 20 s. Superimposed images of a free-falling artificial seed of 1.5 cm size before (**c**) and after (**d**) UV illumination. Photographs of natural (**e**) and artificial (**f**) Javan cucumber seed. Reproduced with permission[50]. Copyright 2024, Springer Nature. All rights reserved. Photographs of the parachute (**g**) and LCN-membrane parachute (**h**). Photographs of natural (**i**) and artificial (**j**) dandelion seed. Photographs of artificial Javan cucumber seed deforms on the left (**k**) and right (**l**) sides induced by UV illumination. **m** The different trajectories of the artificial Javan cucumber seed

gliding in the air. Insets are top-view photographs of the different light-induced wing configurations for each trajectory. The side view photographs of an LCN-membrane parachute show a flat shape originally (**n**) and a bent shape (**o**) under UV illumination. **p** Change of the bending angle $\Delta\alpha$ of the LCN-membrane parachute and corresponding $V_T$, upon different doses of UV irradiation. The error bars are displayed as mean values +/- standard deviation ($n = 3$). The same sample was measured repeatedly. **q** Schematic drawings of an artificial dandelion operating by using UV and weak white light. **r** The snapshot images of artificial dandelion that closes upon UV and opens to take off on top of a constant wind flow upon 50 mW cm$^{-2}$ white light illumination. Wind flow: 100 cm s$^{-1}$. UV illumination: 150 mW cm$^{-2}$, 20 s. All the scale bars are 5 mm.

that have evolved parachute-like structures (such as the dandelion). Herein, we present three types of artificial structures inspired by these modes, details about fabrication see Supplementary Fig. 20–22 and the Supplementary Methods, all leveraging the advantages of azo-LCN. The first structure mimics the design of the Javan cucumber seed, depicted in Fig. 5e, f, its center of gravity is positioned at the middle of the wing, enabling it to glide straight in the air under the assistance of gravity[44]. By exposing different parts of the structure to UV light, the wing can bend either on the left or right (Fig. 5k, i), thereby altering the glide trajectory accordingly (Fig. 5m). The second structure is inspired by the parachute design, as depicted in Fig. 5g, h, this structure comprises three azo-LCN strips with transparent triangular membrane wings attached to them. Under UV light illumination, the middle strips bend, leading the entire structure to close (Fig. 5n, o), consequently altering the terminal velocity (superimposed images, see in Supplementary Fig. 23). As shown in Fig. 5p, increasing the duration of UV exposure results in an increase in its terminal velocity, with the maximum velocity 60% higher than that of the initial velocity (when the bending angle is zero). The third one is artificial dandelion, as depicted in Fig. 5i, j, which is constructed by symmetrically gluing two sets of pappus (18 pieces of filament each) around an azo-LCN film. As schematically illustrated in Fig. 5q, upon exposure to UV light, the artificial dandelion closes its structure, remaining grounded under wind flow during night-time or darkness. However, during daytime or in the presence of visible light, the structure opens and takes off with the assistance of upward wind. As shown in Fig. 5r, it slowly opens under $50\,\text{mW cm}^{-2}$ of white light (half the intensity of sunlight) and completes the take-off process with the assistance of an upward wind speed of $100\,\text{cm s}^{-1}$.

## Discussion

Many microrobotic studies utilize materials that are deformed electrically and magnetically. Here, we emphasize the advantages of the utilization of light fields. Light, i.e., the laser, can propagate over long distances without significant dissipation. Furthermore, remotely directed actuation using light allows for sequestering the power source from the microrobot by eliminating the need for onboard electronics. This offers significant advantages for lightweight the overall structure while still allowing for modulation of its motility from a large stand-off distance. A particular challenge in the realm of sub-gram scale robotics is achieving sustained flight. State-of-the-art approaches rely on onboard electrical power to drive flapping-wing architectures, where advances in miniaturization of drive electronics and wing kinematics have enabled a range of biomimetic designs[16,45,46]. Intriguingly, photoactive liquid crystalline polymers have been shown to demonstrate actuation cycles that include flapping-like motion in the $10^0$-$10^1\,\text{Hz}$ scales[19,47].

The lift force ($F_l$) scales as: $F_l \propto C_L \rho U^2 S_f$, where $C_L$ is the lift coefficient (typically ~ of the order of $10^{-1}$), $\rho$ is the density of air (~ $1.2\,\text{kg m}^{-3}$), $U$ is the wing tip velocity and $S_f$ is the area of the flapping wing[48]. For the centimeter-scale wingspans considered here, $S_f \sim 10^{-4}\,\text{m}^2$. At the minimum, to hover, the $F_l$ must overcome the weight, which for the structures considered here is ~ $10^{-4}\,\text{N}$. The frequency of actuation needed to hover is typically in the $>10^2\,\text{Hz}$ regime. The frequencies at which photo-responsive materials actuate by bending/flapping are at least an order of magnitude slower. Given this infeasibility, an intriguing possibility is demonstrated in this study, where photo-response is used to direct the dynamics of a passive flyer. In this case, the role of light is not to generate lift but to modulate the flight control surfaces via photomechanical morphing.

The first-order kinetics of photoisomerization is associated with the saturation of photostrains with prolonged illumination with UV (trans-cis) or their erasure with visible light (cis-trans) in the azo-LCN. The kinetics of the progressive photostrain generation are intensity-dependent, as shown in Fig. 2c, i. The photoresponse in the material

systems explored here is robust (Supplementary Fig. 12). Supplementary Fig. 24 shows the deformation of the azo-LCN strip upon UV and blue light irradiation, where during the relaxation, bending occurs at about 100° per second. These time scales are within that needed for modulating the responses during the gliding of the microfilters. To further enhance gliding performance, such as reducing terminal velocity, future research may consider employing lightweight materials, e.g. porous LCN[49], for constructing both actuators and passive elements. The ability to optimize the response profiles by modulating the balance between drag and the influence of gravity on the payload offers a platform to broadly tune the trade-space of photo-tunable aerodynamics.

Recently, many passive flying robots with different design configurations have emerged. Supplementary Table 2 summarizes the key factors in recent publications[27,28,31,33]. Research foci vary across disciplines: Advances in microrobotics drive integrated systems for wireless control, sensing, and communication. However, advances in responsive materials offer multimodal actuation profiles using unconventional stimuli such as light. The ability to photo adapt the aerodynamics of microfibers presents a convergence, where the photo strains are sufficient to induce a significant change in aerodynamic properties without constituting a burden in the payload or onboard power.

In conclusion, we have developed an artificial maple samara by assembling a wing-shaped light-responsive film and an additive construct for optimization of mass distribution. This structure exhibits similar airborne autorotating behavior as the natural maple samara. Through external light excitation of ultraviolet and visible, the aerodynamic structure can be reversibly altered via photochemical deformation in LCN-based soft actuator. Placing the artificial seed in a vertical wind tunnel enables observation of reversible changes in altitude and spinning rate controlled by light. We demonstrate that the light can be used to modulate the dispersion of the passive fliers in both indoor and real-world environments. The utilization of photoresponsive materials is also shown to unlock a design space where the fliers can be miniaturized across length scales that span an order of magnitude. Thus, enabling a platform where flight dynamics can be contactlessly tuned. These results present a motif for reimagining how swarms of passive fliers can be controlled by using light.

## Methods
### Materials
4-Methoxybenzoic acid 4-(6-acryloyloxyhexyloxy)phenyl ester (99%, RM 105), 4[4[6-Acryloyloxyhex-1-yl)oxyphenyl]carboxybenzonitrile (99%, RM 23), diacrylate crosslinker 1,4-Bis-[4-(6-acryloyloxyhexyloxy)benzoyloxy]-2-methylbenzene (99%, RM 82) and 4,4′-Bis[9-(acryloyloxy) nonyloxy]azobenzene (95%, ST 04181) were purchased from SYN-THON Chemicals. Bis(2,4,6-trimethylbenzoyl)-phenylphosphine oxide (97%) was purchased from SIGMA Aldrich. pH indicator strips were purchased from MQuant. Reflective Tape was purchased from 3 M. Kapton film was purchased from Chemplex. All chemicals were used as received. Maple samaras and dandelion seeds were collected from plants growing in Tampere (61° 29′ 53″ N, 23° 45′ 36″ E) in October and July 2022, respectively.

### Fabrication of azo-LCN film
To prepare a liquid crystal cell, two coated glass substrates were glued together. One substrate was coated with polyvinyl alcohol (PVA, 5 wt% in water, 3000 rpm, 1 min) and baked at 90 °C for 10 min, then unidirectionally rubbed to attain uniaxial LC alignment. The other substrate was coated with polyimide (JSR OPTMER, 3000 rpm, 1 min), then baked at 180 °C for 20 min to achieve homeotropic alignment. To determine the thickness of the LCN film, a gap between the two glass substrates was introduced by using 20 µm microspheres (Thermo Scientific). The LC mixture containing

52 mol% 4-Methoxybenzoic acid 4-(6-acryloyloxyhexyloxy)phenyl ester, 18 mol% 4[4[6-Acryloxyhex-1-yl)oxyphenyl]carboxybenzonitrile, 21 mol% diacrylate crosslinker 1,4-Bis-[4-(6-acryloyloxyhexyloxy)benzoyloxy]-2-methylbenzene, 6 mol% 4,4'-Bis[9-(acryloyloxy) nonyloxy]azobenzene and 1.5 mol% of photoinitiator were melted at 90 °C and infiltrated into the cell via capillary at 90 °C. After cooling to 50 °C (1 °C min⁻¹), the cell was irradiated with light (420 nm, 50 mW cm⁻², 30 min) for polymerization. Finally, the cell was opened with a razor blade.

## Setup of wind tunnel

The wind tunnel was constructed by using a custom-made cardboard tube, securely connected using adhesive. A fan (San Ace 52, 9GA0512P7A001) was positioned at the base, responsible for generating airflow. The fan received its electrical power from a power supply unit (PSP-2010, Yleiselektroniikka), and the wind speed was regulated through an Arduino Uno control system. To regulate the output airflow, meshes and a flow honeycomb (MAF Airflow Straightener Screen, 76 mm) were put above the wind tunnel.

## Fabrication of the artificial seed

The azo-LCN film (homeotropic alignment in the upper layer, uniaxial alignment in the down layer) was trimmed into the shape of a natural maple samara wing. A natural maple samara root was then glued to the base of the wings. Subsequently, an LCN strip (polydomain, unable to morph in shape) was attached to the upper part of the wing by using glue. Length of azo-LCN wing: 2.5 cm, LCN strip dimension: 1.8 cm × 0.26 cm × 0.02 cm, the mass of the seed root is 24 mg, the mass of the LCN strip is 1 mg, and the mass of LCN wing is 7.1 mg.

## Data availability

The raw data generated in this study have been deposited in Fairdata IDA online storage space (https://ida.fairdata.fi/s/NOT_FOR_PUBLICATION_SW6y2r8db9rt). Data is available from authors on request.

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

## Acknowledgements

H. Z. gratefully acknowledges the financial support of the Academy of Finland (Academy Research Fellow, No. 340263; Mobility, No. 349260; Flagship program, PREIN) and the European Research Council (Starting Grant project ONLINE, No. 101076207). J. Y. gratefully acknowledges funding from the China Scholarship Council (CSC). M. R. S. acknowledges support from the National Science Foundation (1921842).

## Author contributions

H. Z. conceived the idea and supervised the project; J. Y. performed all experiments. J. Y. and H. Z. wrote the manuscript with inputs from M. R. S.

## Competing interests

The authors declare no competing interests.
