## [Peer Review File · Nature Communications]

Photochemically responsive polymer films enable tunable gliding flightsREVIEWER COMMENTS

Reviewer #1 (Remarks to the Author):

This paper presents an interesting design where the authors engineer a light deformable glider that have optically tunable rotation in air. Specifically, it can change the rotational speed of the micro-glider by changing the illumination between ultraviolet and visible light (blue).

While interesting, a significant major revision would be required for this paper to be appropriate for nature communications. Specifically, the authors should address the following important questions.

1) I do not understand the motivation for altering between UV and blue light to change the rotational speed of a micro-glider. Who is projecting the UV light? Once the micro-gliders are released from say a drone, it would not be possible to control what light is been projected on the glider. So how can this research direction be helpful to change the shape of the micro-glider in mid-air? I can imagine applying the Blue or UV light at the drone before releasing the glider, but then it would be a one-time operation right at release and a similar behavior can be achieved by have different shaped micro-gliders.

2) Looks like the UV light and Blue light are illuminated for 10-30 s. That is a rather large duration or something that is falling in the wind as a glider might be in air for around the same duration. In comparison recent work on origami microfliers [1] can change between shapes within tens of milliseconds.

3) The difference in the velocity before and after UV is 98 cm/s and 88 cm/s. This difference is small. From a dispersal perspective how much does it change the dispersal when deployed at say 20-40 m above ground in moderate wind conditions? Explaining this will further strengthen the motivation. A related comment is that the gliders seem to be released in controled setting above just 2 m in Fig. 4 and the separation in Fig. 4c was 15cm. I think it would benefit to understand deployment in real-world settings where wind direction/speed changes with altitude, and the height of the drone is much larger.

4) Fig. 4. d,e,f, claims to have integrated some sensors. Is there a microcontroller? Is there wireless communication? How are you powering these sensors? If not, what is the point of these figures and the claim that you integrated with sensors? This is misleading without power, computing and/or wireless connectivity. If the authors did not build the whole sensor with power and compute, I think this claim should be removed from the abstract/intro.

5) Fig. 2 i,j What are the two colors?

6) Probably the bigger issue that this submission has is that recent work on origami microfliers [Science Robotics'23] demosntrates a end-to-end shape-changing micro-flier that is powered with solar power and was deployed in real-world outdoor deployment and comes with Bluetooth, on-board control,

compute and actuation. I think the authors should have qualitative comparisons with this recent concurrent work.

Similarly in light of [1], the following lines have to be modified since they are no longer true.

Lines 43-44: However, this necessitates the utilization of piezoelectric and dielectric materials that rely on electrical energy transfer, requiring either cables or onboard batteries (the origami work does not require cables or onboard batteries)

Lines 70-72: Thus, there are still many questions opening: Is there an effective actuation method to control the deformation of the glider in the sky? Can deformation induce an impact on the aerodynamics? Can sensors be integrated onboard? (the origami work addresses all these questions positively)

7) The authors say "However, the fragility of the pappus of dandelion limits its load-carrying capacity 26,27 , posing challenges for certain propagations that require transporting heavier payloads or operating in harsh environments" I think more evidence is required for this statement since citation 27 uses light to change shape of a dandelion microflier. Is there evidence that citation 27 cannot carry as much payload and function as the design presented in this work?

[1] Johnson, Kyle & Arroyos, Vicente & Ferran, Amélie & Villanueva, Raul & Yin, Dennis & Elberier, Tilboon & Aliseda, Alberto & Fuller, Sawyer & Iyer, Vikram & Gollakota, Shyamnath. (2023). Solar-powered shape-changing origami microfliers. *Science Robotics*. 8. 10.1126/scirobotics.adg4276.

Reviewer #2 (Remarks to the Author):

In this manuscript, the authors developed shape-changing seed-inspired microfliers by integrating light-responsive LCN wings, enabling reversibly optical tuning of terminal velocity, rotational rate and circling position. The wind dispersal capabilities of the microfliers can also be reversibly tuned under different light illuminations. Furthermore, different environment-sensitive materials are integrated on the microflier to realize different distributed sensing functionalities, such as optical tracking, humidity sensing and pH measurement. The following issues should be well addressed in the form of a major revision, before the work can be considered for publication in *Nature Communications*.

Major Comments:

1. On page 4, the authors divided the seeds into parachuters and gliders according to their flight mechanisms. However, the aerodynamic properties and motion modes of the glider-like and helicopter-like flight are distinct. The authors might want to divide the seeds into three representative types: parachuters, gliders and helicopters (i.e., the microfliers in this work).
2. On page 4, the authors mentioned that “there are still many questions opening: Is there an effective actuation method to control the deformation of the glider in the sky? Can deformation induce an impact on the aerodynamics? Can sensors be integrated onboard?” For the first question, a recent work (Science Robotics 2023, 8, eadg4276) proposed solar-powered shape-changing origami microfliers. For the third question, there have been many works demonstrating multiple sensors integrated with the microfliers, such as pressure sensor, pH sensor, UV sensor, among others. In short, the authors are suggested to change the descriptions about the existing problems to better address the innovation of this paper.
3. In Figure 3h, five cycles of actuation might not be sufficient in many scenarios. The authors are suggested to add more cyclic results and corresponding discussions.
4. In Figure 2j, the relationship between wing deflection angle (α) and light dose is demonstrated. However, the negative values of α (as shown in Movie 2) are not included in this image. Besides, the results and discussions of flight properties (e.g., V_T and D_F) when $\alpha < 0$ should be provided.
5. For actual application scenarios in nature environments, the utilization of UV light stimulus seems impractical. The authors are suggested to add corresponding discussions.
6. The ultimate goal of bionics is to transcend nature. In Figure 1c, d and 4a, the aerodynamic properties of artificial seeds are weaker than natural ones. Are there some effective strategies to improve the capabilities of artificial seeds? The authors are suggested to add corresponding results and discussions through some possible ways, such as introducing complex deformation modes (e.g., twisting) in the LCN wings.

Minor issues:

1. In Figure 2i, j and Figure 3e, f, the y axial titles are not centered.
2. The images in Supplementary Figure 10a and Figure 4a are the same. Please check it.

Reviewer #3 (Remarks to the Author):

Review of the paper “Light-tuned autorotation in seed-inspired, wind-dispersed polymer films”.

In this manuscript, the authors describe how a photosensitive, deformable polymer film mounted together with a seed root, can modify the speed of descendance after illumination. The work is inspired by natural seeds that contain a wing and start to spin around a vertical axis when they fall towards the ground, which helps them to travel over a larger distance when there is a horizontal air flow. The artificial photosensitive structure also rotates when falling down. The experiments involve the observation of the rotation speed, descendance speed and inclination angle of the natural and artificial maple seed, the position of rotation above a vertical air flow and the lateral distance travelled under a horizontal wind speed.

The principle of photosensitivity of the rotation speed due to deformation is well described, and the experimental results are described in sufficient detail. The change in shape of a photosensitive polymer film based on azo-dyes with hybrid alignment (parallel to the surface on one interface and perpendicular to the surface on the other interface) is well established. The novelty of the manuscript is mainly in mounting this film onto a natural seed root and observing the behavior depending on the state of the film. It turns out that the modification of the rotating seed (and the speed of descendance) is very limited. The illumination with UV light can change the rotation speed and the descending speed by only about 10 to 15%. In addition, the UV or blue dose that is required to switch to the other state is large: 0.24 W/cm² for 30 s for UV and 0.3 W/cm² for 30 s for blue. This means a powerful laser has to be focused on the polymer film for quite some time to obtain a significant modification. The high intensities required strongly limit the applicability of the results. The change in the dispersion distance when falling from a certain height, due to the effect of illumination with UV, is limited. If the aim is simply to maximize the dispersion distance during free fall, then it would be much more efficient to optimize the complex structure of the artificial seed, and there would not be any need for photo-sensitivity. Compared to other papers in the field based on photosensitive polymerized liquid crystal, the advancements presented here are limited, therefore I think the paper will be of limited interest to the broad audience of Nature Communications. The submission to a more specialized journal seems appropriate.

In addition, there are some remarks on specific topics.

1. There is a problem with the English in the manuscript. Spelling and grammar are often not correct, and sentences are often not well-formulated. The list of remarks is long. Here are some examples that should be corrected:

multiple degree; many questions opening; a fine-tune of aerodynamic property; By adding weight ... enables; the trans isomer decreases; after ceasing the light; lifetime about 300 minutes; a flatten wing plate; the reason behind is; The construction of wind tunnel; the certain height; This yields a fact that; can be repeated for cycles; Color change of color; a payload rang about few tens of mg; Light, i.e. the laser; tune the trajectory over the air.

2. The description in the "Fabrication of the artificial maple seed" is very brief. There are two parts: a wing and a strip. How are they connected? Is there a double-sided tape between them? Are both photosensitive? Are both changing shape? Which side has homeotropic alignment? Figure 2g seems to indicate that the curvature is limited to a specific region and in the 'curved' state the structure shows a corner instead of a constant curvature. Is the director alignment of the strip and the wing homogeneous over the entire area? In which direction is the director? The geometry shown in Figure 2g is not clear (where are the two parts in the wing, and where are they connected?).

3. In figure 1b, the dimensions of the additional strip do not seem to correspond with the dimensions given in the fabrication section.

4. The insets in Figure 2d are very small. They are too small to see and interpret the content.

REVIEWER COMMENTS

Reviewer #1:

This paper presents an interesting design where the authors engineer a light deformable glider that have optically tunable rotation in air. Specifically, it can change the rotational speed of the micro-glider by changing the illumination between ultraviolet and visible light (blue).

Our answer: We thank the reviewer for the positive comments.

While interesting, a significant major revision would be required for this paper to be appropriate for nature communications. Specifically, the authors should address the following important questions.

1) I do not understand the motivation for altering between UV and blue light to change the rotational speed of a micro-glider. Who is projecting the UV light? Once the micro-gliders are released from say a drone, it would not be possible to control what light is been projected on the glider. So how can this research direction be helpful to change the shape of the micro-glider in mid-air? I can imagine applying the Blue or UV light at the drone before releasing the glider, but then it would be a one-time operation right at release and a similar behavior can be achieved by have different shaped micro-gliders.

Our answer: We thank for this very important criticism. The working principle of the azo-LCN actuator film involves morphing in shape upon exposure to UV light and relaxing upon exposure to visible light – a two-way process. However, during mid-air gliding, only a one-way change of the wing configuration is required. This means that only visible light sources such as LEDs, laser beams, or sunlight are needed to induce shape-morphing during the azo-LCN relaxation stage, thus altering the gliding performance. This operational strategy is emphasized in the revised manuscript through additional relevant discussion:

The bistable actuation brings one benefit for the airborne robots: the deformed structure requires only one-shot UV light illumination for shape-morphing, after which it can maintain the fixed geometry during the entire flight until encountering another manipulated light field (visible light) for shape reprogramming.

Furthermore, we conducted light-induced shape changes during the gliding process. Our experimental data clearly demonstrates an increase in descent velocity due to visible light exposure in mid-air conditions. This data is presented in the new Fig. 4a, b, with corresponding discussions provided below,

We first conduct free-fall experiments with artificial seeds irradiated by UV. The UV is illuminated with 20 s to ensure maximal trans-to-cis conversion and saturation of deformation (Fig. 2c). The seeds are placed at an identical release point of 2 m above the ground inside a static air. About 1 m below the releasing point, there is a visible light zone (460 nm, 760 mW cm⁻²), as schematically shown in Fig. 4a. The visible light field relaxes the wing to original

state (cis-to-trans in azo-LCN) and changes descent velocity in the mid-air. Fig. 4b shows vertical trajectories of four free-falling samples. The seeds are first accelerated under gravitational force and reach aerodynamic equilibrium through spinning after half a meter of falling. After entering the light zone, all samples show an enhancement of descent speed. The ΔV_T is about 17% of the V_T of non-UV exposure, which is consistent with previous examination data shown in Fig. 2d-f, j.

Fig. 4: Light-tuned wind-dispersal. (a) Schematic drawings of light induced shape-morphing of artificial seeds in mid-air. (b) Free falling trajectories of artificial seeds across the visible light zone. Light: 460 nm, 760 mW cm⁻². Inset: The measured terminal velocities (V_T) of artificial seeds before (brown bars) and after entering (red bars) the visible light zone. (c) Schematic drawings of the free-fall experiments of gliding flights of artificial seeds with the help of a crosswind indoors. (d) The dispersal distance of natural maple samaras (purple), artificial seeds after the UV irradiation (blue, 150 mW cm⁻², 20 s) and artificial seeds after blue light illumination (red, 300 mW cm⁻², 30 s) upon different loadings. The error bars indicate s.d. for $n = 3$ measurements. Wind speed: 0.93 ± 0.07 m s⁻¹. (e) The landing spot distribution of natural maple samaras (purple triangle), artificial seeds after the UV (blue square, 150 mW cm⁻², 20 s) and artificial seeds with the blue light irradiation (red dot, 300 mW cm⁻², 30 s) on the X-Y plane indoors. Wind speed: 0.93 ± 0.07 m s⁻¹. (f) The landing spot distribution of artificial seeds before (red dot) and after (blue square) UV illumination in outdoor conditions. Height of the building: 14 m, wind speed at the releasing point: 4 ± 0.5 m s⁻¹. UV: 150 mW cm⁻², 20 s. (g) Statistics of landing point distance for artificial seeds after UV irradiation (red) and the same seeds before the UV irradiation (blue). UV: 150 mW cm⁻², 20 s. Day (h) and night (i) photos of artificial seeds equipped with light reflectors. All scale bars are 1 cm.

Moreover, the photochemical actuation in azo-LCN offers significant advantages for gliding flight, particularly in terms of energy efficiency. We believe this serves as the primary motivation behind our research direction. In the revised version, we have included additional discussion paragraphs to elaborate on this aspect.

Many micro robotic studies utilize materials that are deformed electrically and magnetically. Here, we emphasize the advantages of the utilization of light field. Light, *i.e.*, the laser, can propagate over long distance without significant dissipation. Furthermore, remotely directed actuation using light allows for sequestering the power source from the microrobot by eliminating the need for on-board electronics. This offers significant advantages for light weighting the overall structure while still allowing for modulation of its motility from large stand-off distance. A particular challenge in the realm of sub-gram scale robotics is achieving sustained flight. State-of-the-art approaches rely on on-board electrical power to drive flapping-wing architectures, where advances in miniaturization of drive electronics and wing kinematics have enabled a range of biomimetic designs^{16,44,45}. Intriguingly, photoactive liquid crystalline polymers have been shown to demonstrate actuation cycles that include flapping-like motion in the 10^0 - 10^1 Hz scales^{19,46}.

The lift force (F_l) scales as: $F_l \propto C_L \rho U^2 S_f$, where C_L is the lift coefficient (typically \sim of the order of 10^{-1}), ρ is the density of air ($\sim 1.2 \text{ kg m}^{-3}$), U is the wing tip velocity and S_f is the area of the flapping wing⁴⁷. For the centimeter-scale wingspans considered here, $S_f \sim 10^{-4} \text{ m}^2$. At the minimum, to hover, the F_l must overcome the weight, which for the structures considered here is $\sim 10^{-4} \text{ N}$. The frequency of actuation needed to hover is typically in the $> 10^2 \text{ Hz}$ regime. The frequencies at which photo-responsive materials actuate by bending/flapping are at least an order of magnitude slower. Given this infeasibility, an intriguing possibility is demonstrated in this study, where photo-response is used to direct the dynamics of a passive flyer. In this case, the role of light is not to generate lift but to modulate the flight control surfaces via photomechanical morphing.

Reviewer: 2) Looks like the UV light and Blue light are illuminated for 10-30 s. That is a rather large duration or something that is falling in the wind as a glider might be in air for around the same duration. In comparison recent work on origami microfliers [1] can change between shapes within tens of milliseconds.

Our answer: Thanks for the important question. Prolonged UV/Visible light exposure is to ensure complete isomerization of azobenzene within the material. This process is to achieve distinct deformations between the two stages, facilitating comparison of gliding performances, and enabling complete cis-trans transition for spectral characterization. In the revised version, we have included experimental data demonstrating the rapid response of azo-LCN to blue light excitation.

The first order kinetics of photoisomerization is associated with the saturation of photostrains with prolonged illumination with UV (trans-cis) or their erasure with visible light (cis-trans) in the azo-LCN. The kinetics of the progressive photostrain generation is intensity dependent, as shown in Fig. 2c, i. The photoresponse in the material systems explored here is robust (Supplementary Fig. 12). Supplementary Fig. 24 shows deformation of the azo-LCN strip upon UV and blue light irradiation, where during the relaxation, bending occurs at about 100° per second. These time-scales are within that needed for modulating the responses during the gliding of the microfliers.

Supplementary Fig. 24 | Photo-response of azo-LCN. (a) UV–Vis spectra of a solid actuator film upon UV (150 mW cm⁻²) and 460 nm (300 mW cm⁻²) illumination. Film thickness: 5 μm. (b) Spectral details at the moment upon switching on 460 nm illumination. (c) Deformation kinetics of the azo-LCN strip under UV (150 mW cm⁻²) and 460 nm (300 mW cm⁻²) illumination. Strip thickness: 20 μm. (d) The zoomed-in view of angle change of azo-LCN strip upon switching on 460 nm illumination. Insets: photographs of the actuator strip at the deformed and relaxed stages. The scale bars are 5 mm.

Reviewer: 3) The difference in the velocity before and after UV is 98 cm/s and 88 cm/s. This difference is small. From a dispersal perspective how much does it change the dispersal when deployed at say 20-40 m above ground in moderate wind conditions? Explaining this will further strengthen the motivation.

A related comment is that the gliders seem to be released in controlled setting above just 2 m in Fig. 4 and the separation in Fig. 4c was 15cm. I think it would benefit to understand deployment in real-world settings where wind direction/speed changes with altitude, and the height of the drone is much larger.

Our answer: Thanks for these highly valuable questions. In our opinion, the descent velocity variation ranging from around 100 to 85 cm/s (~15%), while not large, is significant. Firstly, it offers a versatile method for reconfiguring wing shape reversibly. Secondly, the tuning process is continuous, allowing for adjustment to any bending angle, rather than functioning as an ON-OFF event.

Furthermore, the change in descent velocity is highly dependent on the flight mode. In the revised manuscript, we expanded the concept to other gliding modes. In one such mode, i.e., the parachute mode, the descent speed can be reduced by 40% after light-induced deformation, as shown in the new Figure 5 g, h, p.

Fig. 5: Scaling down and conceptual generalization. (a) Photographs of artificial seed with sizes of 3, 1.5, 0.75 and 0.3 cm. (b) The measured terminal velocity (V_T) and the change of terminal velocity (ΔV_T) before and after UV illumination in the seeds with different sizes. UV illumination: 150 mW cm⁻², 20 s. Superimposed images of a free-falling artificial seed of 1.5 cm size before (c) and after (d) UV illumination. Photographs of natural (e) and artificial (f) Javan cucumber seed. Photographs of parachute (g) and LCN-membrane parachute (h). Photographs of natural (i) and artificial (j) dandelion seed. Photographs of artificial Javan cucumber seed deforms on the left (k) and right (l) sides induced by UV illumination. (m) The different trajectories of the artificial Javan cucumber seed gliding in the air. Insets are top view photographs of the different light-induced wing configurations for each trajectory. The side view photographs of an LCN-membrane parachute showing flat shape originally (n) and a bent shape (o) under UV illumination. (p) Change of the bending angle $\Delta\alpha$ of the LCN-membrane parachute and corresponding V_T , upon different doses of UV irradiation. (q) Schematic drawings of an artificial dandelion operating by using UV and weak white light. (r) The snapshot images of artificial dandelion that closes upon UV and opens to take off on top of a constant wind flow upon 50 mW cm⁻² white light illumination. Wind flow: 100 cm s⁻¹. UV illumination: 150 mW cm⁻², 20 s. All the scale bars are 5 mm.

For deployment in real-world conditions, we conducted wind-assisted dispersal by releasing the samples from a height of 14 meters above the ground. The mean separation distance observed was 3 meters, as shown in the new Fig. 4f, g.

(f) The landing spot distribution of artificial seeds before (red dot) and after (blue square) UV illumination in outdoor conditions. Height of the building: 14 m, wind speed at the releasing point: $4 \pm 0.5 \text{ m s}^{-1}$. UV: 150 mW cm^{-2} , 20 s. (g) Statistics of landing point distance for artificial seeds after UV irradiation (red) and the same seeds before the UV irradiation (blue). UV: 150 mW cm^{-2} , 20 s.

We also added explanation into the text,

Thirdly, we conduct multiple free-fall experiments of artificial seeds in outdoor conditions. The seeds are released from a ladder outside of a building, which is 14 m above the ground. The instantaneous wind speed is measured to be about 4 m s^{-1} at the releasing point. A photograph of the experimental location is shown in Supplementary Fig. 15. Fig. 4f shows that the unilluminated artificial seeds land at a horizontal distance about 7 m away from the releasing spot. The UV illuminated samples can travel with an increased distance. Thus, a 3 m mean difference is observed between two sets of samples based on statistics from 30 experiments (Fig. 4g, Supplementary Fig. 16).

Reviewer: 4) Fig. 4. d,e,f, claims to have integrated some sensors. Is there a microcontroller? Is there wireless communication? How are you powering these sensors? If not, what is the point of these figures and the claim that you integrated with sensors? This is misleading without power, computing and/or wireless connectivity. If the authors did not build the whole sensor with power and compute, I think this claim should be removed from the abstract/intro.

Our answer: Thanks for these valuable suggestions. Initially, we aimed to demonstrate that responsive polymer films can function as material sensors without requiring electronics. By detecting changes in reflectance and colour, these films can provide information on humidity and pH levels, aiding in environmental monitoring. We acknowledge the reviewer's concern that our previous presentation could be misleading in the context of micro-electronic control and sensor communication. Therefore, we have relocated the sensor information to the Supplementary Information section to explain these additional material functionalities. The text has been revised accordingly:

As shown in Fig. 4d, as the loading weight increases the dispersal distance drops,

indicating the ability to modulate the aerodynamics, while sustaining a payload up to tens of mg. To enable their tracking, we incorporated the artificial seed with an optical indicator. The indicator film can reflect light directionally with large portion of back reflection along the reversed direction of incident light. This allowed their utilization in non-laboratory conditions over extended flight paths (Fig. 4h, i). Other stimuli-responsive polymer films, *i.e.*, pH indicator and humidity-responsive polymer, can be also incorporated onto the seed body. This assembly allows for optical monitoring of the environmental conditions by tracking the colors of the responsive elements, which constitute the payload of the microfliers. Details can be found in Supplementary Fig. 17, while the sample preparation of humidity-responsive polymer is outlined in Supplementary Fig. 18 and the Supplementary Note.

In abstract.

Secondly, we show that the polymer film geometry can be scaled down to enable miniature gliders with similar light tunability. Thirdly, we extend the material platform to other three gliding modes, *i.e.*, Javan cucumber seed-like glider with steerable direction, LCN-membrane parachute capable of 40% change in terminal velocity and artificial dandelion seed for self-dispersal upon weak light. The findings offer new opportunities for achieving distributed microflier swarms, whose flight dynamics can be controlled contactlessly.

Reviewer: 5) Fig. 2 i,j What are the two colors?

Our answer: We have modified the figure to highlight the content.

Reviewer: 6) Probably the bigger issue that this submission has is that recent work on origami microfliers [Science Robotics'23] demonstrates a end-to-end shape-changing micro-flier that is powered with solar power and was deployed in real-world outdoor deployment and comes with Bluetooth, on-board control, compute and actuation. I think the authors should have qualitative comparisons with this recent concurrent work.

Our answer: Thanks for your excellent comments and for mentioning one of the most important papers in the field, [Solar-powered shape-changing origami microfliers. Science Robotics].

The concept demonstrated in this paper has a different focus, it is important to note that the weight and size differ from our approach. We have compared the key parameters and added a supplementary table (please see below) in the revised version to highlight these differences.

Here, we would like to express our views on the cross-disciplinary research. The device described in [Solar-powered shape-changing origami microfliers. Science Robotics] represents a remarkable achievement in robotic design, seamlessly integrating an electromagnetic actuator for shape-morphing actuation, a polyimide film for the parachute body, sensors, solar cell panels for electric energy generation, circuits including Bluetooth and antenna, and soft electronic connections. The prototype addresses practical challenges faced by roboticists, such as controlling descent behavior from tumbling to stability, self-powering, and Bluetooth-enabled sensing and control. Our manuscript adopts a materials science perspective, to

conduct research that is driven by curiosity and future challenges. Our efforts aim to explore new possibilities enabled by these simple polymer films in future research.

We believe that, whether viewed from a roboticist's perspective (focused on precise control in control, sensing, and communication) or from a material scientist's viewpoint (considering a very simple piece of polymer film and the diversity of actuation mode), both approaches ultimately encounter the same technical challenge crucial for experiments in both flights. Our study and the approach taken in the [Science Robotics] article demonstrate facts that both gliding structures are passive fliers, and both require only minimal geometry deformation to affect their gliding performances. Instead of relying on complicated electronic materials and control mechanisms, can material scientists offer simple material strategies to achieve such deformation? In the revised manuscript we try to provide our opinion about this point. We have included a discussion to illustrate this point.

Recently, many passive flying robots with different design configurations have emerged. Supplementary Table 1 summarizes the key factors in recent publications^{26,27,30,32}. Research foci vary across disciplines: Advances in microrobotics drive integrated systems for wireless control, sensing and communication. However, advances in responsive materials offer multimodal actuation profiles using unconventional stimuli such as light. The ability to photoadapt the aerodynamics of microfliers present a convergence, where the photostrains are sufficient to induce a significant change in aerodynamic properties, without constituting a burden in the payload or on-board power.

	Weight (mg)	Terminal velocity (cm s ⁻¹)	Actuation mechanism	Control in air	Continuous shape-morphing	Response time
Origami microfliers ¹	414	140~ 180	Solar electricity and electromagnetism	Yes	No	25 ms
Artificial dandelion ²	4	41.0~ 98.0	Light-heat induced gas desorption	Yes	Yes	2.5 s
Rotary flight ³	5.3	72.9~ 80.0	Light-heat induced gas desorption	No	No	650 ms
Fliers in this study						
Artificial maple samara	32.1	86.8~ 98.4	Photochemical effect	Yes	Yes	< 3 s
Artificial glider	24.8	60.0~ 88.3	Photochemical effect	Yes	Yes	< 3 s

Parachute	9.2	70.7~ 115.4	Photochemical effect	Yes	Yes	< 3 s
Artificial dandelion seed	1.2	45.8~ 81.2	Photochemical effect	Yes	Yes	< 3 s

Supplementary Table 1 | Comparison of microfliers based on passive flight mode.

Reviewer: Similarly in light of [1], the following lines have to be modified since they are no longer true.

Lines 43-44: However, this necessitates the utilization of piezoelectric and dielectric materials that rely on electrical energy transfer, requiring either cables or onboard batteries (the origami work does not require cables or onboard batteries).

Our answer: The introduction has been divided into two parts: active fliers and passive fliers (gliders). The discussion of piezoelectric and dielectric actuators is specifically related to active flying modes. We have modified the text to illustrate this in a better way.

To achieve a hovering flying robot (**active flight mode**) with dimensions in the centimeter...

In the natural kingdom, many seeds have evolved to utilize wind-assisted **passive flight** mechanisms effectively, ...

Reviewer: Lines 70-72: Thus, there are still many questions opening: Is there an effective actuation method to control the deformation of the glider in the sky? Can deformation induce an impact on the aerodynamics? Can sensors be integrated onboard? (the origami work addresses all these questions positively).

Our answer: Thanks for all the valuable comments, which allow us to re-think our illustration of research motivation. In the revised version, we have modified the research questions as,

A recent development has successfully addressed several important issues in gliding robots³², *i.e.*, fast actuation, self-powering, control of the geometry mid-flight to modulate the aerodynamics, while accessing functionalities using on-board sensors and wireless communication. Such approach is based on a complex integration of multiple electronic elements within a miniaturized origami parachute. Alternately, optical shape morphing polymers set up an intriguing opportunity for controlling responsiveness from stand-off distances, contactlessly. Thus, setting up the question in the context of microfliers: Can light be a direct way to reconfigure the wing geometry and provide dynamic control of the gliding performance? Can a single piece of polymer be used to modulate responsiveness, instead of an integrated electromechanical system built on multiple materials? Furthermore, can this shape-morphing allow for facile tuning of the gliding to elicit a range of modalities and abilities to steer the trajectory?

Here, we attempt to explore the aforementioned scientific questions by reporting light-

deformable polymers that exhibit optically tuning in gliding performances in the air.

Reviewer: 7) The authors say "However, the fragility of the pappus of dandelion limits its load-carrying capacity 26,27, posing challenges for certain propagations that require transporting heavier payloads or operating in harsh environments" I think more evidence is required for this statement since citation 27 uses light to change shape of a dandelion microflier. Is there evidence that citation 27 cannot carry as much payload and function as the design presented in this work?

Our answer: Thanks for the interesting comments. We have performed additional experiments to demonstrate the differences in load-carrying capacity between the maple samara mode and the dandelion seed mode. The corresponding data can be found in Supplementary Fig. 1.

For a quantitative comparison of load-carrying between helicopter-like and parachute-like fliers, see in Supplementary Fig.1.

Supplementary Fig. 1 | Comparison of the load-carrying capacities. Photographs of natural maple samaras (a) and artificial seeds (b). Photographs of natural (c) and artificial (d) dandelion seeds. (e) The changes of terminated velocity ΔV_T of natural maple samaras and artificial seed, natural and artificial dandelion seeds along addition of weight. All the scale bars are 5 mm.

Reviewer #2:

In this manuscript, the authors developed shape-changing seed-inspired microfliers by integrating light-responsive LCN wings, enabling reversibly optical tuning of terminal velocity, rotational rate and circling position. The wind dispersal capabilities of the microfliers can also be reversibly tuned under different light illuminations. Furthermore, different environment-sensitive materials are integrated on the microflier to realize different distributed sensing functionalities, such as optical tracking, humidity sensing and pH measurement. The following issues should be well addressed in the form of a major revision, before the work can be considered for publication in Nature Communications.

Our answer: We thank the reviewer for the positive comments.

Reviewer: Major Comments:

1. On page 4, the authors divided the seeds into parachuters and gliders according to their flight mechanisms. However, the aerodynamic properties and motion modes of the glider-like and helicopter-like flight are distinct. The authors might want to divide the seeds into three representative types: parachuters, gliders and helicopters (i.e., the microfliers in this work).

Our answer: Thanks for this constructive suggestion. We have made the modifications to the text accordingly.

These mechanisms can be roughly divided into **three categories**²²: (1) Seeds with a parachute-like structure, such as dandelions, have a filament structure that creates air resistance; (2) Seeds **that can glide in the air**, such as Javan cucumber that have wing-like structures, **and (3) maple samara and similar species** can autorotate in the air, generating lift through a helicopter-like mechanism.

Reviewer: 2. On page 4, the authors mentioned that “there are still many questions opening: Is there an effective actuation method to control the deformation of the glider in the sky? Can deformation induce an impact on the aerodynamics? Can sensors be integrated onboard?” For the first question, a recent work (Science Robotics 2023, 8, eadg4276) proposed solar-powered shape-changing origami microfliers. For the third question, there have been many works demonstrating multiple sensors integrated with the microfliers, such as pressure sensor, pH sensor, UV sensor, among others. In short, the authors are suggested to change the descriptions about the existing problems to better address the innovation of this paper.

Our answer: Thanks for this valuable comment. We have modified the text to explicitly describe the research questions from our perspective. They are,

Thus, setting up the question in the context of microfliers: Can light be a direct way to reconfigure the wing geometry and provide dynamic control of the gliding performance? Can a single piece of polymer be used to modulate responsiveness, instead of an integrated electromechanical system built on multiple materials? Furthermore, can this shape-morphing allow for facile tuning of the gliding to elicit a range of modalities and abilities to steer the trajectory?

For detailed comparison with (Science Robotics 2023, 8, eadg4276), please see in the response to Reviewer #1, Comment no.6.

Reviewer: 3. In Figure 3h, five cycles of actuation might not be sufficient in many scenarios. The authors are suggested to add more cyclic results and corresponding discussions.

Our answer: Thanks for raising this important issue. In the revised version, we have included experimental data on 25 cycles of spinning performance and 100 cycles of strip deformation. Both sets of data have shown no observable decay. The data can be found in the new Supplementary Figure 12.

Supplementary Fig. 12 shows the data of 25 cyclic light tunings of gliding performance inside the wind tunnel, and 100 cycles of photomechanical deformations in an actuator strip. Within experimental time span, no decay in material capability is observed.

Supplementary Fig. 12 | The cycle test of artificial seed. (a) The change of h (red line) and Ω (blue dot) of artificial seed upon altering the UV and visible excitations in 25 cycles. UV: 360 nm, 150 mW cm⁻²; visible light: 460 nm, 400 mW cm⁻². (b) The snapshot images of an artificial seed with reversible height change by altering the illumination. (c) Deformation of an azo-LCN strip during a hundred light actuation cycles. UV: 360 nm, 150 mW cm⁻²; visible light: 460 nm, 400 mW cm⁻². (d) Photographs of light-induced deformation of an azo-LCN strip. All scale bars are 5 mm.

Reviewer: 4. In Figure 2j, the relationship between wing deflection angle (α) and light dose is demonstrated. However, the negative values of α (as shown in Movie 2) are not included in this image. Besides, the results and discussions of flight properties (e.g., V_T and D_F) when $\alpha < 0$ should be provided.

Our answer: Thanks for these important comments. We have conducted additional experiment, and provided the data in new Fig. 2 j, k.

Upon further increase in UV irradiation above $30 \times 10^3 \text{ J m}^{-2}$, the film bends to the opposite side with a negative α , and meanwhile both V_T and Ω start increasing.

Fig. 2: Photochemical actuation and light tuning in artificial seeds. (a) Chemical composition of the photomechanical actuator. (b) UV-Vis spectra of a solid actuator film. Thickness of the film: $5 \mu\text{m}$. Alignment: splayed. (c) The change of trans-isomer content in the azo-LCN film upon irradiation of different UV intensities. Superimposed images of a free-falling artificial seed as initial state (d), after UV (240 mW cm^{-2} , 12 s) illumination (e) and then after blue light (300 mW cm^{-2} , 20 s) irradiation (f). The shape of the artificial seed wing before (g) and after (h) UV irradiation. UV: 240 mW cm^{-2} , 12 s. (i) Deformation kinetics of the azo-LCN film under different UV intensities, insets: photographs of azo-LCN strip bending after UV illumination. (j) Change of terminal velocity (V_T , red ring) and spinning rate (Ω , blue dot) of the artificial seed after receiving different UV doses. (k) Change of the deflection angle α (red ring) of the wing and corresponding descent factor (D_F , blue dot) upon different doses of UV irradiation. The error bars indicate s.d. for $n = 3$ measurements. UV wavelength: 360 nm. After each experiment, the film is illuminated by blue light (460 nm , 300 mW cm^{-2} , 30 s) for complete shape recovery. All scale bars are 5 mm.

Reviewer: 5. For actual application scenarios in nature environments, the utilization of UV light stimulus seems impractical. The authors are suggested to add corresponding discussions.

Our answer: Thanks for raising this very important comment. And thank you for highlighting this very important comment. Reviewer #1 also provided similar criticism. Herein, we kindly guide the Reviewer to refer to the response provided to Reviewer

#1, comments no. 1 & 2, as well as the answers provided to all Reviewers, point no. 2.

Reviewer: 6. The ultimate goal of bionics is to transcend nature. In Figure 1c, d and 4a, the aerodynamic properties of artificial seeds are weaker than natural ones. Are there some effective strategies to improve the capabilities of artificial seeds? The authors are suggested to add corresponding results and discussions through some possible ways, such as introducing complex deformation modes (e.g., twisting) in the LCN wings.

Our answer: Thanks for raising these excellent comments. In our view, our artificial structures surpass nature in three key aspects. Firstly, they enable the continuous and reversible tuning of gliding through light, whereas natural seeds typically remain static or respond very slowly to environmental stimuli such as light and humidity. Secondly, our concept is scalable, allowing for adaptation to smaller sizes. Thirdly, it can be extended to other gliding modes, such as those inspired by dandelions, Javan cucumber seeds, and parachutes.

We have added one new Figure 5 to demonstrate the downscaling and generalization concepts.

The photochemical actuation in azo-LCN offers an opportunity to miniaturize the flier. As shown in Fig. 5a, we fabricated different sizes (3, 1.5, 0.75 and 0.3 cm) using a laser cutter. All sizes of artificial seeds achieve mid-air autorotation with their descent speeds averaging around 100 cm s^{-1} . As depicted in Fig. 5b, after exposure to UV light, the terminal velocity of all sizes of artificial seeds decreased due to a reduction in the deflection angle α of the wing. Superimposed images of the light tunable seeds during the free-falling are shown in Fig. 5c, d and Supplementary Fig. 19. The results are consistent with that observed in Figure 2.

The shape-morphing azo-LCN also provides the opportunity to extend the concept to various passive flight modes. In nature, alongside those capable of autorotation like the maple samara, there exist seeds with sail-like wings for gliding (e.g., Javan cucumber) and seeds that have evolved parachute-like structures (such as the dandelion). Herein, we present three types of artificial structures inspired by these modes, details about fabrication see Supplementary Fig. 20-22 and the Supplementary Note, all leveraging the advantages of azo-LCN. The first structure mimics the design of the Javan cucumber seed, depicted in Fig. 5e, f, its center of gravity is positioned at the middle of the wing, enabling it to glide straight in the air under the assistance of gravity⁴³. By exposing different parts of the structure to UV light, the wing can bend either on the left or right (Fig. 5k, i), thereby altering the glide trajectory accordingly (Fig. 5m). The second structure is inspired by the parachute design, as depicted in Fig. 5g, h, this structure comprises three azo-LCN strips with transparent triangular membrane wings attached to them. Under UV light illumination, the middle strips bend, leading the entire structure to close (Fig. 5n, o), consequently altering the terminal velocity (superimposed images, see in Supplementary Fig. 23). As shown in Fig. 5p, increasing the duration of UV exposure results in an increase in its terminal velocity, with the maximum velocity 60% higher than that of the initial velocity (when the bending angle is zero). The third one is artificial dandelion, as depicted in Fig. 5i, j, which is constructed by symmetrically gluing two sets of pappus (18 pieces of filament each) around an azo-LCN film. As schematically illustrated in Fig. 5q, upon exposure to UV light, the artificial dandelion closes its structure, remaining grounded under wind flow during night-time or darkness. However, during daytime or in the presence of visible light, the structure opens and takes off with the assistance of upward wind. As shown in Fig. 5r, it slowly opens under 50 mW cm^{-2} of white light (half the intensity of

sunlight) and completes the take-off process with the assistance of an upward wind speed of 100 cm s^{-1} .

Fig. 5: Scaling down and conceptual generalization. (a) Photographs of artificial seed with sizes of 3, 1.5, 0.75 and 0.3 cm. (b) The measured terminal velocity (V_T) and the change of terminal velocity (ΔV_T) before and after UV illumination in the seeds with different sizes. UV illumination: 150 mW cm^{-2} , 20 s. Superimposed images of a free-falling artificial seed of 1.5 cm size before (c) and after (d) UV illumination. Photographs of natural (e) and artificial (f) Javan cucumber seed. Photographs of parachute (g) and LCN-membrane parachute (h). Photographs of natural (i) and artificial (j) dandelion seed. Photographs of artificial Javan cucumber seed deforms on the left (k) and right (l) sides induced by UV illumination. (m) The different trajectories of the artificial Javan cucumber seed gliding in the air. Insets are top view photographs of the different light-induced wing configurations for each trajectory. The side view photographs of an LCN-membrane parachute showing flat shape originally (n) and a bent shape (o) under UV illumination. (p) Change of the bending angle $\Delta\alpha$ of the LCN-membrane parachute and corresponding V_T upon different doses of UV irradiation. (q) Schematic

drawings of an artificial dandelion operating by using UV and weak white light. (r) The snapshot images of artificial dandelion that closes upon UV and opens to take off on top of a constant wind flow upon 50 mW cm^{-2} white light illumination. Wind flow: 100 cm s^{-1} . UV illumination: 150 mW cm^{-2} , 20 s. All the scale bars are 5 mm.

About further improve the capacity of the artificial spinning seed, we have included the following discussion:

To further enhance gliding performance, such as reducing terminal velocity, future research may consider employing lightweight materials, e.g. porous LCN⁴⁸, for constructing both actuators and passive elements.

Reviewer: Minor issues:

1. In Figure 2i, j and Figure 3e, f, the y axial titles are not centered.

Our answer: We have modified the figures accordingly.

Reviewer: 2. The images in Supplementary Figure 10a and Figure 4a are the same. Please check it.

Our answer: We apologize for the mistake. The figures are modified accordingly.

Reviewer #3:

Review of the paper “Light-tuned autorotation in seed-inspired, wind-dispersed polymer films”.

In this manuscript, the authors describe how a photosensitive, deformable polymer film mounted together with a seed root, can modify the speed of descendance after illumination. The work is inspired by natural seeds that contain a wing and start to spin around a vertical axis when they fall towards the ground, which helps them to travel over a larger distance when there is a horizontal air flow. The artificial photosensitive structure also rotates when falling down. The experiments involve the observation of the rotation speed, descendance speed and inclination angle of the natural and artificial maple seed, the position of rotation above a vertical air flow and the lateral distance travelled under a horizontal wind speed. The principle of photosensitivity of the rotation speed due to deformation is well described, and the experimental results are described in sufficient detail. The change in shape of a photosensitive polymer film based on azo-dyes with hybrid alignment (parallel to the surface on one interface and perpendicular to the surface on the other interface) is well established. The novelty of the manuscript is mainly in mounting this film onto a natural seed root and observing the behavior depending on the state of the film. It turns out that the modification of the rotating seed (and the speed of descendance) is very limited. The illumination with UV light can change the rotation speed and the descending speed by only about 10 to 15%.

Our answer: Thank you for these important comments. We acknowledge that it was our oversight to neglect the technical details and manuscript structure, resulting in potential confusion among readers and a lack of clarity regarding the manuscript's novelties. In the revised version, we have reorganized the contents to clearly highlight the following novelties of the manuscript: 1) The ability to tune gliding properties using light while in mid-air. 2) The capacity for downsizing using the same material platform. 3) The potential for generalization to other gliding modes, such as Javan cucumber seed, parachute, and dandelion seed. Notably, the parachute-like structure can achieve a 40% change in descent velocity. New Figures 4 and 5 have been included in the revised version to visually depict these novelties.

Furthermore, we believe that a ~15% change in descent velocity in the artificial seed glider, although not substantial, is significant for several reasons. Firstly, it offers an unconstrained method to reconfigure the wing shape reversibly. Secondly, the tuning is continuous, allowing for adjustment to any bending angle, rather than being limited to an ON-OFF event. Thirdly, our study represents a successful demonstration of such tuning in mid-air conditions.

Reviewer: In addition, the UV or blue dose that is required to switch to the other state is large: 0.24 W/cm² for 30 s for UV and 0.3 W/cm² for 30 s for blue. This means a powerful laser has to be focused on the polymer film for quite some time to obtain a significant modification. The high intensities required strongly limit the applicability of the results.

Our answer: We thank you for these valuable comments. Again, it was our mistake that led to this misunderstanding. Allow us to briefly explain: the prolonged exposure

to UV/visible light is to ensure complete isomerization of azobenzene within the material. This is to achieve distinct deformation between the two stages and complete the cis-trans transition for spectral characterization. In the revised version, we have included experimental data demonstrating the rapid response of LCN to blue light excitation.

The first order kinetics of photoisomerization is associated with the saturation of photostrains with prolonged illumination with UV (trans-cis) or their erasure with visible light (cis-trans) in the azo-LCN. The kinetics of the progressive photostrain generation is intensity dependent, as shown in Fig. 2c, i. The photoresponse in the material systems explored here is robust (Supplementary Fig. 12). Supplementary Fig. 24 shows deformation of the azo-LCN strip upon UV and blue light irradiation, where during the relaxation, bending occurs at about 100° per second. These time-scales are within that needed for modulating the responses during the gliding of the microfliers.

Supplementary Fig. 24 | Photo-response of azo-LCN. (a) UV-Vis spectra of a solid actuator film upon UV (150 mW cm^{-2}) and 460 nm (300 mW cm^{-2}) illumination. Film thickness: $5 \mu\text{m}$. (b) Spectral details at the moment upon switching on 460 nm illumination. (c) Deformation kinetics of the an azo-LCN strip under UV (150 mW cm^{-2}) and 460 nm (300 mW cm^{-2}) illumination. Strip thickness: $20 \mu\text{m}$. (d) The zoomed-in view of angle change of azo-LCN strip upon switching on 460 nm illumination. Insets: photographs of the an actuator strip at the deformed and relaxed stages. The scale bars are 5 mm.

Worth noting that we did not utilize any lasers in this study; all light illuminations were provided by LED sources or lamps. Furthermore, intensities of a few hundred mW cm^{-2} are only a few times stronger than sunlight (100 mW cm^{-2}), and thus should not be considered high intensities. To exemplify low-intensity performance, we conducted additional experiments on the light tuning of gliding mid-air using 760 mW cm^{-2} blue light irradiation, and the self-opening of a dandelion glider under 50 mW cm^{-2} white light illumination. The results are shown in new Figure 4a, b and 5q, r.

Fig. 4: Light-tuned wind-dispersal. (a) Schematic drawings of light induced shape-morphing of artificial seed in mid-air. (b) Free falling trajectories of artificial seeds across the visible light zone. Light: 460 nm, 760 mW cm⁻². Inset: The measured terminal velocities (V_T) of artificial seeds before (brown bars) and after entering (red bars) the visible light zone. (c) Schematic drawings of the free-fall experiments of gliding flights of artificial seeds with the help of a crosswind indoors. (d) The dispersal distance of natural maple samaras (purple), artificial seeds after the UV irradiation (blue, 150 mW cm⁻², 20 s) and artificial seeds after blue light illumination (red, 300 mW cm⁻², 30 s) upon different loadings. The error bars indicate s.d. for $n = 3$ measurements. Wind speed: 0.93 ± 0.07 m s⁻¹. (e) The landing spot distribution of natural maple samara (purple triangle), artificial seeds after the UV (blue square, 150 mW cm⁻², 20 s) and artificial seeds with the blue light irradiation (red dot, 300 mW cm⁻², 30 s) on the X-Y plane indoors. Wind speed: 0.93 ± 0.07 m s⁻¹. (f) The landing spot distribution of artificial seeds before (red dot) and after (blue square) UV illumination in outdoor conditions. Height of the building: 14 m, wind speed at the releasing point: 4 ± 0.5 m s⁻¹. UV: 150 mW cm⁻², 20 s. (g) Statistics of landing point distance for artificial seeds after UV irradiation (red) and the same seeds before the UV irradiation (blue). UV: 150 mW cm⁻², 20 s. Day (h) and night (i) photos of artificial seeds equipped with light reflectors. All scale bars are 1 cm.

Fig. 5: Scaling down and conceptual generalization. (a) Photographs of artificial seed with sizes of 3, 1.5, 0.75 and 0.3 cm. (b) The measured terminal velocity (V_T) and the change of terminal velocity (ΔV_T) before and after UV illumination in the seeds with different sizes. UV illumination: 150 mW cm⁻², 20 s. Superimposed images of a free-falling artificial seed of 1.5 cm size before (c) and after (d) UV illumination. Photographs of natural (e) and artificial (f) Javan cucumber seed. Photographs of parachute (g) and LCN-membrane parachute (h). Photographs of natural (i) and artificial (j) dandelion seed. Photographs of artificial Javan cucumber seed deforms on the left (k) and right (l) sides induced by UV illumination. (m) The different trajectories of the artificial Javan cucumber seed gliding in the air. Insets are top view photographs of the different light-induced wing configurations for each trajectory. The side view photographs of an LCN-membrane parachute showing flat shape originally (n) and a bent shape (o) under UV illumination. (p) Change of the bending angle $\Delta\alpha$ of the LCN-membrane parachute and corresponding V_T upon different doses of UV irradiation. (q) Schematic drawings of an artificial dandelion operating by using UV and weak white light. (r) The snapshot images of artificial dandelion that closes upon UV and opens to take off on top of a constant wind flow upon 50 mW cm⁻² white light illumination. Wind flow: 100 cm s⁻¹. UV illumination: 150 mW cm⁻², 20 s. All the scale bars are 5 mm.

Reviewer: The change in the dispersion distance when falling from a certain height, due to the effect of illumination with UV, is limited. If the aim is simply to maximize the dispersion distance during free fall, then it would be much more efficient to optimize the complex structure of the artificial seed, and there would not be any need for photo-sensitivity. Compared to other papers in the field based on photosensitive polymerized liquid crystal, the advancements presented here are limited, therefore I think the paper will be of limited interest to the broad audience of Nature Communications. The submission to a more specialized journal seems appropriate.

Our answer: We thank for the comments. We believe the research motivation is effectively communicated in the revised version. For details regarding the manuscript novelties, please refer to the Responses to All Reviewers section.

Additionally, we would like to highlight the following merits achieved in this revised manuscript, which have not been reported in the literature:

- i) Photomechanical tuning of gliding behavior in mid-air (new Figure 4).
- ii) Ability to easily scale down autorotated gliders (new Figure 5).
- iii) Versatility of the material platform for use in other glider modes, as demonstrated by the Javan cucumber seed, parachute, and dandelion seed examples (new Figure 5).

Reviewer: In addition, there are some remarks on specific topics.

1. There is a problem with the English in the manuscript. Spelling and grammar are often not correct, and sentences are often not well-formulated. The list of remarks is long. Here are some examples that should be corrected:

multiple degree; many questions opening; a fine-tune of aerodynamic property; By adding weight ... enables; the trans isomer decreases; after ceasing the light; lifetime about 300 minutes; a flatten wing plate; the reason behind is; The construction of wind tunnel; the certain height; This yields a fact that; can be repeated for cycles; Color change of color; a payload rang about few tens of mg; Light, i.e. the laser; tune the trajectory over the air.

Our answer: We thank for the comments. We have made every effort to improve the English language quality. In this revised version, while it may not be native-level, we believe the writing does not pose any obstacles to the communication of scientific ideas.

Reviewer: 2. The description in the "Fabrication of the artificial maple seed" is very brief. There are two parts: a wing and a strip. How are they connected? Is there a double-sided tape between them? Are both photosensitive? Are both changing shape? Which side has homeotropic alignment? Figure 2g seems to indicate that the curvature is limited to a specific region and in the 'curved' state the structure shows a corner instead of a constant curvature. Is the director alignment of the strip and the wing homogeneous over the entire area? In which direction is the director? The geometry shown in Figure 2g is not clear (where are the two parts in the wing, and where are they connected?).

Our answer: Thank you for these important comments. We have provided a comprehensive description of the sample fabrication, along with Supplementary Fig. 8 to aid in illustration:

After exposure to UV light, the deflection angle α of the wing decreases, as depicted by the side view photographs of artificial seed in Fig. 2g, h, and the tip displacement (d) data in Fig. 2i. Top view photographs of artificial seed before and after photo-actuation, see in Supplementary Fig. 8.

Supplementary Fig. 8 | The shape changes of the artificial seed wing. Top view photographs of artificial seed before (a) and after (b) UV illumination. UV: 240 mW cm⁻², 12 s. The scale bars are 5 mm.

Method.

Fabrication of the artificial seed. The azo-LCN film (homeotropic alignment in upper layer, uniaxial alignment in down layer) was trimmed into the shape of natural maple samara wing. A natural maple samara root was then glued to the base of the wing. Subsequently, an LCN strip (polydomain, unable to morph in shape) was attached to the upper part of the wing by using glue. Length of azo-LCN wing: 2.5 cm, LCN strip dimension: 1.8 cm × 0.26 cm × 0.02 cm, the mass of the seed root is 24 mg, the mass of the LCN strip is 1 mg and the mass of LCN wing is 7.1 mg.

Reviewer: 3. In figure 1b, the dimensions of the additional strip do not seem to correspond with the dimensions given in the fabrication section.

Our answer: We apologize about the mistake. The figure is corrected accordingly.

Reviewer: 4. The insets in Figure 2d are very small. They are too small to see and interpret the content.

Our answer: We thank for the comment. The figure is modified accordingly.

REVIEWERS' COMMENTS

Reviewer #1 (Remarks to the Author):

The authors have addressed all my comments. I appreciate them for thoroughly addressing the issues in the previous submission.

I think the paper is ready for publication and would be of interest to the community.

Reviewer #2 (Remarks to the Author):

I have read carefully the authors' responses to all reviewers as well as the revised manuscript. In my opinion, the authors have made substantial modifications to address the comments from all reviewers. The overall quality of the paper has been improved considerably by taking the inputs from the reviewers. As such, I feel the paper is now ready for publication in Nature Communications.